# Surface structure of water from soft X-ray second harmonic generation

David J. Hoffman [1] ✉, Shane W. Devlin[2,3,4], Douglas Garratt [1],
Sasawat Jamnuch[5,6], Jacob A. Spies[2,7], Bailey R. Nebgen [2,7],
Daniel Schacher [4], Alexandria Do[8], Franky Bernal [2,9], Erika J. Riffe [2,9],
Kristjan Kunnus[1], Christina Y. Hampton[1], Joseph Duris [1], David Cesar[1],
Nicholas Sudar[1], Georgi L. Dakovski[1], Walter S. Drisdell [9], Keith V. Lawler [4],
Agostino Marinelli [1], Michael W. Zuerch [2,7], Richard J. Saykally[2,9],
Craig P. Schwartz [4] ✉, Tod A. Pascal [8] ✉ & Jake D. Koralek [1] ✉

The microscopic structure of water's surface is crucial to many natural and industrial processes, but studying its hydrogen bond (H-bond) network directly remains challenging due to the required interfacial sensitivity of experimental techniques. By leveraging advances in flat liquid sheet microjets and terawatt-scale attosecond soft X-ray pulses from the LCLS X-ray free electron laser, we employed soft X-ray second harmonic generation (SXSHG) spectroscopy to examine the liquid water/vapor interface. SXSHG combines the elemental selectivity of X-ray spectroscopies with the surface selectivity of SHG and gives access to the electronic structure of interfacial species. Here, we show the SXSHG spectrum differs from bulk water's X-ray absorption, with its peak shifted several eV, indicating a vastly different electronic environment at the interface as compared to the bulk. First-principles electronic structure calculations show the signal is highly sensitive to H-bond interactions, such as water molecules accepting a single H-bond, which are surface abundant.

Water and its interfaces are ubiquitous in physical, chemical, and biological processes[1–4]. Water's tetrahedral hydrogen bond (H-bond) network leads to the unique macroscopic properties of its liquid phase, such as its anomalously high boiling point and density[5,6], however the H-bond network is necessarily different at the surface due to the lack of H-bonding partners. Due to its fundamental importance, the surface of water has been extensively studied by a wide array of experimental techniques, ranging from measurements of macroscopic properties such as surface tension[7] to probes with finite penetration depth, such as photoemission spectroscopy[8,9], particle scattering[10], and various spectral decomposition and reflectivity-based techniques[11–13]. The

development of nonlinear surface-selective spectroscopies, such as UV-visible second harmonic generation (SHG) and vibrational sum frequency generation (SFG) have provided some structural information about water interfaces. However, the interpretation of the resulting complex spectra is challenging and remains the subject of much debate[14–19]. Therefore, despite the pervasiveness of aqueous interfaces and the many tools developed to study them, there are still substantial open questions about their electronic and molecular structure and how they connect to important macroscopic phenomena.

Soft X-ray second harmonic generation (SXSHG) applies the surface sensitivity of SHG to element-specific X-ray transitions, and can probe

[1]SLAC National Accelerator Laboratory, Menlo Park, CA, USA. [2]Department of Chemistry, University of California, Berkeley, CA, USA. [3]Advanced Light Source, Lawrence Berkeley National Laboratory, Berkeley, CA, USA. [4]Nevada Extreme Conditions Laboratory, University of Nevada, Las Vegas, NV, USA. [5]Theiss Research, La Jolla, CA, USA. [6]Material Measurement Laboratory, National Institute of Standards and Technology (NIST), Gaithersburg, MD, USA. [7]Material Science Division, Lawrence Berkeley National Laboratory, Berkeley, CA, USA. [8]Aiiso Yufeng Li Family Department of Chemical and Nano Engineering, University of California, San Diego, La Jolla, CA, USA. [9]Chemical Sciences Division, Lawrence Berkeley National Laboratory, Berkeley, CA, USA. ✉e-mail: djhoff@slac.stanford.edu; craig.schwartz@unlv.edu; tpascal@ucsd.edu; koralek@slac.stanford.edu

electronic valence states not easily accessible with UV-visible spectroscopies[20–26]. As SXSHG almost-exclusively relies on the intense X-ray pulses produced by X-ray free electron lasers (XFELs), it is a relatively new technique that is still quickly developing. The foundational work on graphite slabs over a range of thicknesses demonstrated the possibility of the measurement and the surface-selectivity of the measurement[24]. A later study on a boron-Parylene N junction showed the sensitivity to buried interfaces as well as the ability to extract intermolecular distances with Angstrom precision[21]. Work on LiNbO$_3$ has also demonstrated the polarization-dependence of the signal as well as element-dependent signatures from the Li K-edge and Nb N-edge[22].

The initial states in soft X-ray spectroscopies like SXSHG comprise tightly bound core electrons, which cause the corresponding transitions to be almost entirely dominated by atom-dependent binding energies that are separated by 10–100 s of eV. In the case of water, soft X-ray absorption spectroscopy (XAS) of the oxygen K-edge is well-established as a sensitive probe of its hydrogen bonding network, and has been the subject of several reviews[27,28]. This sensitivity arises primarily because of the strong transitions from the core oxygen 1 s orbitals and the lowest-lying unoccupied valence orbitals, which have σ* character and are localized on the hydrogen atoms. The water XAS spectrum is then particularly sensitive to the H-bond donation character of the probed water molecules, and suggests that the SXSHG spectrum can provide similar information for the interfacial water molecules.

SXSHG is a nonlinear process, where two soft X-ray photons are absorbed and a new photon with twice the incident energy is emitted (shown schematically in Fig. 1, inset). The resulting SHG signal has a characteristic quadratic intensity dependence with the incident beam's intensity:

$$I_{SHG}(2\omega) = \left|\chi_{eff}^{(2)}(2\omega)\right|^2 (I_0(\omega))^2 \qquad (1)$$

Critically, $\chi_{eff}^{(2)}$, the effective second-order nonlinear susceptibility at a given photon energy, is symmetry forbidden in centrosymmetric media within the dipole approximation[29]. Practically, in the case of liquid water, this means that the only observed SHG signals should arise from interfaces, which inherently break inversion symmetry. Previous experimental and theoretical work on solid samples has demonstrated that 98% of the SXSHG signal is generated from the three atomic layers nearest to the interface[24], and similar probe depths are expected in the case of liquid water.

Despite its purported advantages, several factors make SXSHG experiments on water particularly challenging. First, nonlinear X-ray processes require extreme levels of photon intensity and coherence, which are currently only available from XFELs. In the present work, X-ray pulses were produced using the X-ray Laser-Enhanced Attosecond Pulses (XLEAP) technique, which enables the XFEL to produce sub-femtosecond pulses with greater peak intensities than are possible with the standard spontaneously amplified stimulated emission processes commonly used at XFELs[30]. These pulses were further amplified to TW-scale peak power using superradiant amplification[31]. Second, the strong attenuation of soft X-rays in both air and water requires SXSHG measurements to be performed in a vacuum environment. Here, we employed a flat liquid sheet microjet, which provides a constantly-refreshing liquid target with optically flat liquid interfaces open to the vacuum environment[32–36]. The submicron sheet is thin enough to detect both the fundamental and harmonic signals in transmission geometry, which greatly simplifies the experimental design by allowing for alignment of the spectrometer without the liquid target, minimizing the impacts of possible jet fluctuations, and enabling for the simultaneous measurement of the XAS and SXSHG from the same sample.

With these advances, we measure and report the SXSHG spectrum of the water/vapor interface across the oxygen K-edge (~540 eV). While the measured signal was dominated by background photons from the XFEL source and competing intensity-dependent transient absorption (TA) effects in the water sample, covariance analysis of the harmonic (~1080 eV) and fundamental (~540 eV) regions of the spectra measured at each set photon energy revealed the SXSHG signal and permitted subtraction of the background as a function of intensity-dependence.

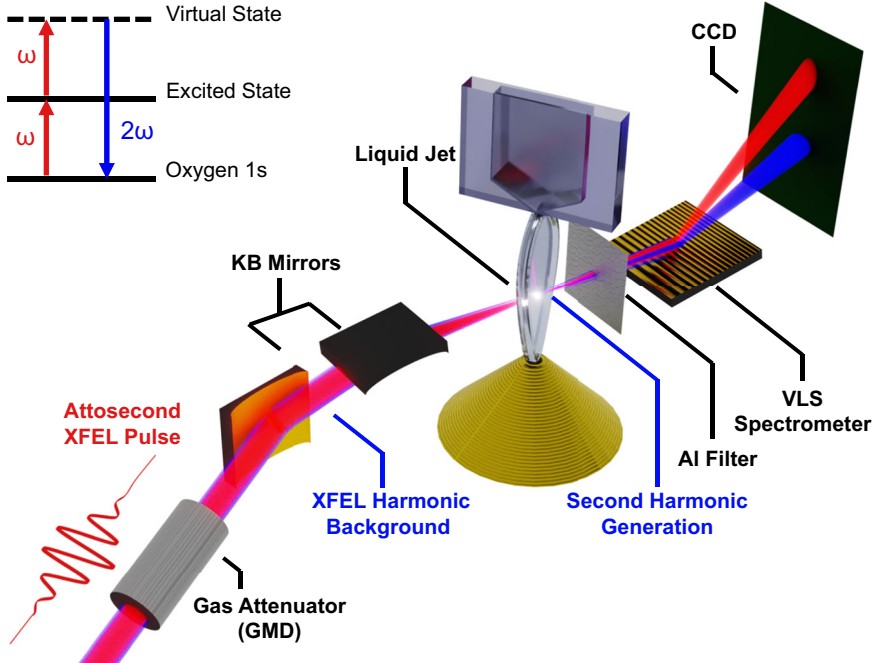

**Fig. 1 | Schematic depiction of the SXSHG experiment.** Attosecond soft X-ray pulses are produced by the XFEL. The pulses intrinsically contain some photons in the second harmonic region. The pulse energy is measured non-invasively by a nitrogen gas attenuator (GMD). The pulse is focused by a pair of Kirkpatrick-Baez mirrors and impinges a water sheet jet at a 70° angle of incidence. The fundamental and harmonic from the XFEL pulse are partially absorbed by the bulk water and SHG is generated at the surfaces. The pulse passes through an aluminum filter to preferentially attenuate the much brighter fundamental. The fundamental and harmonic are spectrally resolved on the detector. Inset: Schematic energy level diagram of the SXSHG process.

The SXSHG signal was only observed above the oxygen K-edge, consistent with resonant enhancement expected from the sample. The peak of the full SXSHG spectrum appeared several eV higher than the peak of the bulk water XAS, differentiating the SXSHG signal from the bulk measurement. These results were supported with first-principles electronic structure calculations, which also captured the measured blue-shift in the SXSHG spectrum compared to the XAS. The simulated SXSHG response of the interfacial water molecules was found to be particularly sensitive to the number of acceptor H-bonds, making this technique a direct probe of the broken H-bond structure of interfacial water analogous to, but distinct from, XAS. Using these calculations, it was determined that the major contributors to the full SXSHG spectrum were water molecules accepting a single H-bond that are significantly more populous at the water surface than in the bulk and agree with state-of-the-art simulations of the liquid water surface.

## Results

### Soft X-ray second harmonic generation spectroscopy

The experimental configuration is shown in a schematic diagram of the SXSHG experiment is shown in Fig. 1 and described in the "Materials and Methods". Briefly, attosecond X-ray pulses with pulse energies of 100 s of µJ were tuned across the oxygen K-edge region from ~520–560 eV in 5 eV steps and were focused using a pair of Kirkpatrick-Baez mirrors to a ~10 µm FWHM spot on the liquid sheet target. The pulse energy was measured non-invasively shot-by-shot using a gas monitor detector (GMD), which we will use as a proxy for intensity throughout this work. For each photon energy, the fundamental and harmonic were measured simultaneously in the transmission geometry on a shot-by-shot basis. The pulses were spectrally resolved to monitor both the harmonic and fundamental spectral regions simultaneously. The XFEL produced a substantial background in the harmonic region that was about four orders of magnitude weaker than the fundamental. Spectral datasets for each XFEL configuration were then taken with (jet-in) and without (jet-out) the liquid sheet in the beam to enable background subtraction of the XFEL harmonic, which additionally enables XAS measurements of the liquid jet across the fundamental region.

The intrinsic shot-to-shot fluctuations of the XFEL pulse energy were used to generate the intensity dependence for identifying nonlinear signals. The pulse energy varies stochastically from <100 to >500 µJ with moderate spectral changes due to shot-by-shot variation in the electron beam[31]. Jet-in and jet-out data were binned by pulse energy and the jet-out data was subtracted from the jet-in data after accounting for absorption through the water sample. An example of intensity-dependent difference signals in the harmonic region are shown in Fig. 2B for a dataset taken at 550 eV (above the oxygen K-edge). The XFEL harmonic and squared fundamental spectra are also shown as blue and red dashed curves, respectively. The squared fundamental spectrum indicates where an SHG signal should appear if $|\chi^{(2)}|^2$ is flat across the spectral region. The complicated behavior of the intensity-dependent spectra was assigned to three distinct processes as illustrated in Fig. 2A, C. Linear absorption and an unexpected TA process that is linear in intensity (Supplementary Fig. 2) both reduce the counts across the entire spectral region, while the SHG increases the counts only in the region that roughly corresponds to the squared fundamental profile. We associate the observed TA features with intrapulse ionization of the water sample, because characteristic water ionization spectral features were also observed in the fundamental region oxygen K-edge XAS[37]. The induced harmonic absorption could then be potentially explained by an increased effective electron density per the Drude model[38].

The decomposition of the difference signal in Fig. 2B into TA and SHG components is supported through the use of well-established covariance mapping techniques[39]. The experimental covariance map for the 550 eV dataset is shown in Fig. 2D and a modeled map is shown in Fig. 2E generated from the jet-out data and the intensity-dependent features illustrated in Fig. 2A. These covariance maps are 2D contour

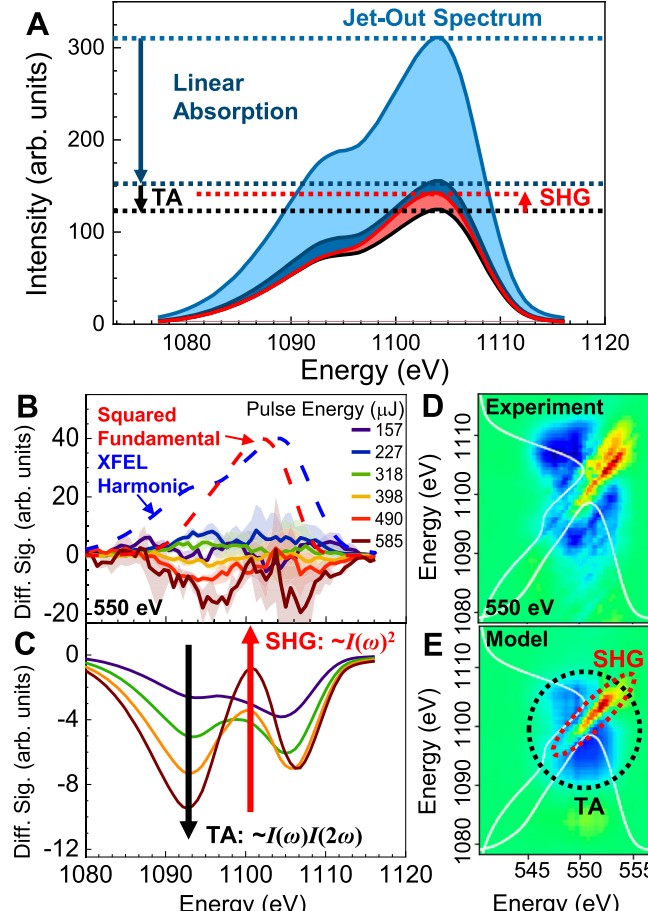

**Fig. 2 | Intensity-dependent processes in the harmonic region. A** The harmonic produced by the XFEL is attenuated by intrinsic linear absorption through the water jet and a transient absorption (TA) induced by the fundamental. The SHG process produces additional counts in the harmonic region. **B** Difference signals for 550 eV at selected binned fundamental pulse energies (µJ). Colored shaded areas represent standard errors for the signal between three datasets. The blue and red dashed curves are the XFEL harmonic and squared fundamental spectra respectively. **C** Depiction of the intensity-dependent difference signals using the features in (**A**), reproducing (**B**). **D** Experimental covariance difference map for 550 eV showing intensity-dependent features. **E** Modeled best-fit covariance map with the intensity-dependent features from (**A**) indicated. The correlated positive SHG feature and negative TA background are apparent. Color bar magnitude is arbitrary but shared.

plots of the covariance of the detector pixels, which indicate how the intensities measured at different photon energies dispersed on the detector vary with respect to each other after background subtraction of intrinsic correlations from the light source. Blue features correspond to a negative correlation (i.e., more fundamental counts correlate to fewer harmonic counts), while red features correspond to a positive correlation (more fundamental counts correlate to more harmonic counts). The x- and y-axes correspond to the fundamental and harmonic regions, respectively, and the white curves along each axis are the corresponding (jet-out) average spectra. The proposed TA signal then results in a broad uncorrelated blue feature, as the TA signal must be nonresonant across the harmonic region and the absorption across the fundamental is similarly flat above the oxygen K-edge, although the measured behavior of the TA depends on the fundamental and harmonic pulse profiles. By contrast, the SHG process will produce a signal that is correlated along the diagonal line corresponding to the harmonic frequency being twice the fundamental frequency.

The best-fit parameters for the TA signals as well as the $|\chi^{(2)}|^2$ spectral response were determined by a simultaneous fitting to the experimental

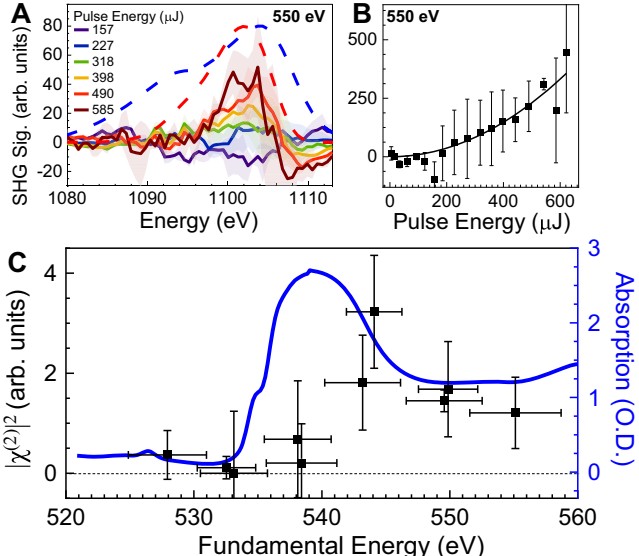

**Fig. 3 | Intensity and spectral dependence of the isolated SXSHG signal.**
**A** Background-subtracted intensity-dependent SHG signals from the data shown in
Fig. 2B. **B** Intensity of the integrated isolated SHG signal vs. the fundamental.
Quadratic fit shown. Error bars represent standard error between three measure-
ments. **C** Full spectral dependence of the nonlinear response $|\chi^{(2)}|^2$ vs. the mean
XFEL photon energy. SHG signal only appears well above the O K-edge from
simultaneously measured XAS (blue curve). Y-error bars from statistical error and
estimated systematic error. X-error bars derived from the FWHM of the XFEL
fundamental spectra.

covariance map and the isolated binned SHG signals (see "Materials and
Methods" for a detailed description). The covariance map (Fig. 2E) pro-
duced from this relatively straightforward model accurately reproduces
the major qualitative features of the experimental data, although it mis-
ses some of the fine structure in both signals. Figure 3A shows the
resulting intensity-dependent isolated SHG signals from the same best fit
model. The isolated signal was positive and increasing with fundamental
intensity, and was in the expected spectral location (red dashed curve).
The integrated intensity-dependent signal was then plotted against the
integrated fundamental (Fig. 3B). The best fit curve (Eq. 1) gives the value
of $|\chi^{(2)}|^2$ and demonstrates the quadratic intensity dependence of the
isolated SHG signal.

The full SXSHG spectrum was then generated by tuning the XFEL
photon energy across the oxygen K-edge in 5 eV steps. The ten col-
lected datasets were all processed in the same manner as described
above, wherein the $|\chi^{(2)}|^2$ response was determined by the simulta-
neous fitting of the covariance maps and the binned isolated SHG
signals. Best fit $|\chi^{(2)}|^2$ parameters are shown for each dataset in Fig. 3C,
plotted against the simultaneously measured XAS of bulk water (blue
curve), illustrating the onset of measured $|\chi^{(2)}|^2$ signal. The y-error bars
in Fig. 3C capture the statistical error as well as estimated systematic
error in modeling the TA background process, detailed in "Materials
and Methods". A clear edge is apparent in the $|\chi^{(2)}|^2$ spectrum within
several eV of the oxygen K-edge measured by XAS. Covariance maps
and difference signals below the K-edge only show evidence of TA
features (in both the harmonic and fundamental spectral regions).
Surprisingly, the onset of the $|\chi^{(2)}|^2$ signal does not occur at the peak of
the XAS spectrum (~540 eV) but is blue-shifted by ~5 eV. The full
dataset then suggests a $|\chi^{(2)}|^2$ spectral response for the water surface
that is distinct from the bulk water XAS.

### First principles simulations
Insights into the origin of the observed SXSHG signal were obtained
using first principles simulations of the SHG response from perturbation

theory within density functional theory (DFT). The methods used are
analogous to those in prior SXSHG studies[20–24,40] (as detailed in the
Materials and Methods), and are briefly outlined here. First, we obtained
ensemble averaged H-bond structures from classical molecular dynam-
ics (MD) simulations using the many-body polarizable water model (MB-
pol)[41], which has been rigorously validated to simultaneously capture
both the bulk[42] and interfacial[43] physics of water at ambient conditions.
We then determined the water mass-density distribution as a function of
distance from the instantaneous water/vapor interface (Supplementary
Fig. 6B)[44], and quantified the H-bond characteristics of interfacial mole-
cules by calculating the potential of mean force (Supplementary
Fig. 6C)[45]. The populations of different major species as distinguished by
number of H-bond acceptors and donors (each ranging from zero to
two) are shown in Fig. 4A and illustrated in Fig. 4B. SXSHG spectra were
calculated for each unique H-bond configuration (Fig. 4C, minor con-
figurations can be found in Supplementary Fig. 7D), sampled from an MD
simulation of a smaller simulation cell for computational feasibility. The
full simulated signal is generated on a per-atom basis, allowing for the
contributions from particular molecules to be isolated.

The simulated SXSHG spectra were found to be particularly sensi-
tive to the number of acceptor H-bonds. This stands in contrast to linear
XAS measurements, which have been shown to be sensitive to the
degree of donor H-bonds due to the low-lying water valence orbitals
being localized on the hydrogen atoms, but not particularly sensitive to
the number of acceptor H-bonds[28]. In particular, interfacial single
acceptor/no donor species ($A_1D_0$, blue trace in Fig. 4C) presents several
SXSHG features at ~532 eV, 540 eV and a particularly intense main peak at
544 eV. Conversely, the calculated spectrum of the $A_2D_2$ species (red
trace in Fig. 4C), the dominant constituent of bulk water, is largely fea-
tureless over the calculated energy range. The spectra of each H-bond
donor/acceptor species were then weighted based on the interfacial
population thermodynamics at 277 K to match the expected experi-
mental temperature due to evaporative cooling in vacuum[46] to obtain a
computational SXSHG spectrum. Figure 4E shows that the simulated
spectrum is in excellent agreement with the experimental measure-
ments. Additional simulations also show the SXSHG signal results from
broken symmetry at the interface, as the corresponding SXSHG
response of a bulk water system was found to be 50x less intense even
with the limited simulation cell size (Supplementary Fig. 7A). These cal-
culations inherently include bulk-active quadrupole contributions as
well (see "Materials and Methods"). Combined with previous experi-
mental and theoretical work on other samples showing thickness
independence[24] and polarization-dependence[22] consistent with dipole
selection rules, these simulations support our claim of surface selectivity.

### Discussion
The experimentally observed blue shift of ~5 eV in the surface SXSHG
spectrum relative to the bulk XAS was also captured by our first-
principles SXSHG simulations (Supplementary Fig. 7C), but these
simulations do not provide an orbital-level description. To better
understand this effect, we performed high-level quantum chemistry
calculations of the natural transition orbitals resulting from the over-
lap of the 1s core electron with the various SXSHG excited states
(corresponding to excitonic virtual states and the process depicted by
the blue arrow in the schematic SHG energy level diagram in Fig. 1).
These calculations, performed in the single molecule limit, revealed a
manifold of core-excited states at twice 539 eV (sx1), which are weakly
dipole allowed, and another manifold of states at twice 543 eV (sx2)
with much stronger oscillator strengths. Simulations of $A_xD_y$ species
and their first solvation shell waters revealed similar manifolds for all
configurations, with larger oscillator strength correlating with smaller
exciton size[47] within this energy range. These manifolds and corre-
sponding molecular orbitals for an isolated molecule are shown in
Fig. 4D (gray trace) relative to the major water molecule valence
transitions (red trace). Note that the gas phase XAS appears at lower

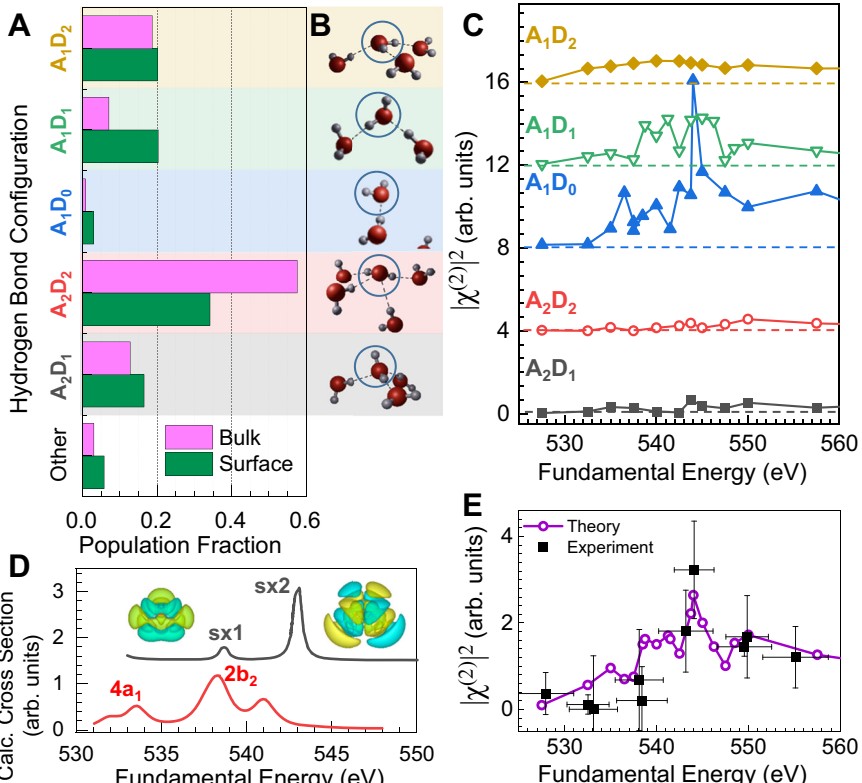

**Fig. 4 | First-principles simulations of SXSHG spectra. A** Population analysis of water molecules in the bulk (purple) and first interfacial layer (green), grouped by number of acceptor (A) and donor (D) H-bonds from MD calculations. **B** Schematic depictions of the major water H-bond configurations. **C** Simulated SXSHG spectra for different H-bond configurations that contribute significantly to the total spectrum. Intensities are all to scale, vertically offset for clarity (dashed lines indicate respective zeros). **D** TD-DFT calculations of the excited states of an isolated water molecule. Valence states are shown in red and highly excited SXSHG states are shown in gray, plotted at half energy. The SXSHG manifolds (orbital isosurfaces inset) are blue shifted relative to the valence states. **E** The resulting simulated spectrum (purple) is compared to experiment (black) from Fig. 3C. The population in (**A**) are used to weight the spectra of the various H-bonding configurations in (**C**).

energy than the liquid phase XAS[27,28]. The half-energy sx1 and sx2 states are also several eV blue-shifted relative to the valence transitions, which is suggestive of an orbital overlap explanation for this effect. However, these calculations do not consider the transition between the valence and sx1/sx2 manifolds. The sx2 manifold of states was found to have intramolecular charge transfer character, as determined by the transition charge metric[48], with large oscillator strengths due to strong dipole induced-dipole couplings. This manifold of states for both the single molecule and clusters is also particularly sensitive to molecular asymmetry (Supplementary Fig. 8), which is known to occur in liquid water in the presence of broken H-bonding[49,50]. These states may provide a partial explanation for the sensitivity to acceptor H-bonds as well, as the $4a_1$ state localized on the hydrogen atoms have poor spectral overlap with the sx1/sx2 manifolds, and potentially better overlap with higher energy orbitals with more oxygen character.

Our MD simulations revealed an interface that is decorated with broken H-bond species. The single-acceptor ($A_1D_x$) species were both greatly enhanced at the surface relative to the bulk and produce the largest signal per molecule. The single-acceptor-single-donor ($A_1D_1$, green trace in Fig. 4C) species make up ~20% of the surface molecules with three times their bulk prevalence, and it is the primary contributor to the 544 eV main peak and a post-edge peak in the SXSHG spectrum at 550 eV. The single acceptor, no donor ($A_1D_0$, blue trace in Fig. 4C) species represents ~3% of the surface water molecules (four times their bulk prevalence) but has an outsized influence on the main SXSHG peak at 544 eV and the smaller peak at 536.5 eV.

Notably, the SXSHG spectrum of the tetrahedral double-acceptor-double-donor species ($A_2D_2$, red trace in Fig. 4C) has only a weak response at 550 eV due to the high degree of symmetry, and while this species is the majority component of the bulk, about half as many (~35%) of the surface waters adopt this configuration. In general, the weak response from the more symmetric species necessitates information from other more established techniques to fully describe the liquid surface, such as vibrational SFG. The weak SXSHG activity of the $A_2D_2$ species provides an interesting contrast to the vibrational SFG data, as the OH stretch of individual species provide a strong response in the SFG which is minimized in the final spectrum due to phase cancellations due to the opposing orientations of molecules near the surface[19,51–53]. Here, the per-species response is reduced, possibly due to the valence molecular orbitals gaining more s-character when tetrahedrally bonded, as has been seen in simulations of the water XAS spectrum[54]. This may reduce either the one or two-photon cross-section in the symmetric $A_2D_2$ species.

Thus, when considering the calculated spectra and population analysis, we conclude that the main peak in the full SXSHG spectrum encodes the signature of broken, single acceptor H-bonding. This result corroborates the sensitivity of using oxygen 1 s electronic transitions to probe hydrogen bonding with high selectivity, and shows that SXSHG can complement optical and infrared nonlinear methods in characterizing the water surface to enable a full understanding of the liquid surface.

In summary, we demonstrated SXSHG as a probe of the water/vapor interface by combining intense attosecond XFEL pulses with flat liquid sheet microjets. The experimental spectrum was interpreted by complementary first-principles electronic structure calculations of water molecules in distinct H-bond environments, and weighting the simulated spectra by the prevalence of each H-bond configuration from extensive MD simulations. Overall, we find that the major

contributors to the full SXSHG spectrum are water molecules accepting a single H-bond, which are enhanced at the interface in agreement with state-of-the-art MD simulations. This sensitivity to H-bond acceptors was an unexpected result, as the bulk XAS is known to be more sensitive to H-bond donors.

While in this current work, the conclusions that we are able to draw are limited due to the low signal-to-noise of the SXSHG feature above the XFEL background, there are a number of strategies that could be pursued to improve the measurement. Selective absorptive and spatial filters can be employed to preferentially remove the XFEL harmonic before reaching the sample target, or a reflective geometry could be employed to minimize the harmonic background by utilizing Brewster's angle. Target thickness, angle-of-incidence, and X-ray polarization dependence could also be examined in future work to experimentally demonstrate surface selectivity.

Most significantly, the commissioning of new high repetition rate XFELs such as LCLS-II will enable the collection of SXSHG data at tens of kHz instead of the 120 Hz used in this study. We expect this to help evaluate and validate the various water models/potentials used in the literature for simulating the vapor/water interface. In addition to future measurements on neat water, it will be possible to study a variety of scientifically and industrially interesting systems, including the interfacial response of water due to the presence of small molecules and ions, the electrical double layer, and the solvation environment of biomolecules. Using the orders-of-magnitude statistical improvements provided by next generation XFELs, high quality SXSHG data is expected to provide a rigorous test of theoretical models and further refine our knowledge of the water surface and more complex interfaces and environments.

## Methods
### Experimental setup
The experiments were carried out at the chemRIXS end station of the Linac Coherent Light Source (LCLS) XFEL at SLAC National Accelerator Laboratory. The attosecond pulses were optimized for peak intensity, with several hundred μJ of energy per pulse, bandwidths of ~8 eV, and nominal durations of several hundred attoseconds[31]. Jet-in and jet-out datasets were taken with ~100,000 shots each, corresponding to 15 min of acquisition.

The water target was a liquid sheet that was produced using a microfluidic chip with a converging channel geometry (Micronit Micro 1, flow rate 2.2 mL/min). The liquid sheet is free flowing at high enough velocity (~10 m/s) to ensure a fresh target for each X-ray pulse at 120 Hz. Based on the white-light thin film interference pattern produced by the sheet[55] and the observed XAS, the sheet thickness in the measured region is ~500 nm. To produce SHG from a liquid surface, the electric field of the incident light must have a component normal to the surface[29]. The sheet was rotated to a 70° angle of incidence (relative to sheet normal) with the X-ray pulses having *p*-polarization, giving an effective optical path length of 1.5 μm. Measurements were made in-vacuum with a chamber pressure of ~1 mTorr.

The transmitted pulses were first attenuated using 4.2 μm of aluminum filters which preferentially absorbed the fundamental. They were spectrally resolved using a variable line spacing grating spectrometer in first order (resolving power ~1300) and measured with a CCD detector (Andor Newton_SO) that was able to monitor both the harmonic and fundamental spectral regions simultaneously for each XFEL shot at 120 Hz (Supplementary Fig. 1). There is a broad scattered light background which scales linearly with the pulse energy, so an estimated scalar background value on a shot-by-shot basis was determined from the average of the high energy side of the detector which was then subtracted from the entire spectrum.

The scattered light background can be seen to also have some oscillatory features which scale with pulse energy. This background is shown in Supplementary Fig. 1C. While the counts from the

background (typically < 5 per pixel) are too small by nearly an order of magnitude to produce the main observed intensity-dependent features in the harmonic region, they likely provide some distortion to the edges of the XFEL harmonic. In addition to the XFEL fundamental and second harmonic, there are smaller peaks around 675 and 875 eV which are smaller scattered fundamental signals. The stronger peak appearing at 675 eV was used to correct for detector saturation in the main fundamental peak on very strong shots. Finally, the second-order harmonic peak (which overlaps the fundamental peak) was estimated from the first-order harmonic peak and subtracted from the fundamental on a shot-by-shot basis, which improves the calculated XAS and covariance maps in the strongly absorbing spectral regions. The attenuation of the harmonic peak through a series of aluminum filters was found to match literature values[56], so there is no major fundamental component to the observed harmonic peak.

Histograms of pulse energies as measured by the GMD for six consecutive runs are shown in Supplementary Fig. 1D, showing the machine behavior is consistent over these timescales. Average spectra of the fundamental and harmonic for each GMD bin are shown in Supplementary Fig. 1E and 1F, respectively. There are moderate changes to the spectrum with pulse energy, especially in the lowest quartile, but the higher energy pulses are fairly consistent. The full analysis described below uses these intensity-dependent spectra in the calculation of the TA and SHG contributions to the signal.

### Intensity-dependent signals analysis
Because of the poor correlation between the XFEL fundamental and harmonic intensities, the only filtering that was done was the removal of dropped XFEL shots from the dataset. For the fundamental intensity-dependent analysis, the data set was binned according to the GMD pulse energy measurement, which is upstream of the interaction point and is equivalent between the jet-in and jet-out datasets[57]. The binned jet-in and jet-out datasets could then be used to analyze intensity-dependent features in both spectral regions.

The full, average spectrum will be notated as $I_0(\Omega)$ and $I(\Omega)$ for the jet-out and jet-in harmonic spectra. The harmonic spectra as binned on the integrated fundamental spectra, $s$, will be given as $I_0(\Omega|s)$ and $I(\Omega|s)$. Similarly, the full fundamental spectra will be given as $I_0(\omega)$ and $I(\omega)$ and the spectra binned on the integrated harmonic spectra, $S$, will be given as $I_0(\omega|S)$ and $I(\omega|S)$.

For calculating the difference signals shown in Fig. 2B of the main text, the binned jet-out data was first scaled by the average transmission through the jet at each photon energy:

$$T(\Omega) = I(\Omega)/I_0(\Omega) \tag{2}$$

Where $T$ represents the transmission at harmonic frequency $\Omega$. $T(\Omega)$ $I_0(\Omega|s)$ was subtracted from each corresponding jet-in dataset $I(\Omega|s)$ to yield the difference signal.

From the model of the harmonic signals presented in Fig. 2A, the observed signal at a given photon energy in the harmonic region depends on the magnitude of the TA signal and the magnitude of the SHG signal:

$$I(\Omega|s) = \left(T_0(\Omega) + M(\Omega)s\right)\left(I_0(\Omega|s) + I_{SHG}(\Omega|s)\right) \tag{3}$$

Where $T_0$ is the linear transmission of the harmonic through the sheet in the absence of any TA effect and $M$ is the magnitude of the TA effect (experimentally found to vary across the harmonic region), and $I_{SHG}$ is the SHG signal at the given frequency and fundamental intensity.

From the observed jet-in and jet-out dataset, the measured SHG signal for a given bin is:

$$I_{SHG}(\Omega|s) = \frac{I(\Omega|s)}{\left(T_0(\Omega) + M(\Omega)s\right)} - I_0(\Omega|s) \tag{4}$$

This form follows from the fact that the observed SHG is generated almost entirely from the sheet surface facing the beam and is therefore attenuated in the same way as XFEL harmonic. As the fundamental is strongly attenuated by the sheet (OD > 1 at all energies we find SHG signal), there is a greater than 100-fold reduction in the SHG signal generated at the back interface. Therefore any SHG produced by the back interface has a negligible contribution on the measured signals. The isolated SHG signal was then integrated across the spectral region for determination of the effective $\chi^{(2)}$ response.

Initial guesses for the TA parameters were generated by using Eq. 3 and assuming $I_{SHG}$ is negligible. The transmission for each GMD bin could then be fit to a line to yield an estimate for $T_0$ and $M$ which were used for the full fitting with the covariance maps. Examples of these linear fits are shown in Supplementary Fig. 2A at the spectral regions shown in Supplementary Fig. 2B. This technique would underestimate the magnitude of the TA response in the presence of an SHG signal, which necessitates the more complex covariance map fitting. Intensity-dependent features are also apparent in the fundamental region. While these are also most likely caused by radiolysis from the fundamental pulse, it was found to be useful for the covariance analysis to correlate the magnitude of the TA features in the fundamental with the integrated XFEL harmonic:

$$I(\omega|S) = (t_0(\omega) + m(\omega)S)I_0(\omega|S) \qquad (5)$$

The values of $t_0$ and $m$ could then be estimated from a linear regression of the data binned to the integrated XFEL harmonic, $S$, as discussed in the previous section.

## Covariance mapping

The covariance mapping[39] visualizes the covariance matrix as a contour plot. For our purposes, we principally care about the correlations between the harmonic and fundamental regions of the detector. Each element of the covariance matrix is the covariance between the measured counts on two pixels. For the jet-out data, the covariance between a pixel in the fundamental region and a pixel in the harmonic region is given by:

$$\mathrm{cov}(I_0(\omega), I_0(\Omega)) = \mathrm{E}[I_0(\omega)I_0(\Omega)] - \mathrm{E}[I_0(\omega)]\mathrm{E}[I_0(\Omega)] \qquad (6)$$

Where E[...] indicates the expectation value. This measurement captures the intrinsic correlations between the fundamental and XFEL harmonic. For the jet-in data, we can use Eqs. 3 and 5 to get:

$$\mathrm{cov}(I(\omega), I(\Omega)) = \mathrm{cov}\left( \begin{array}{c} (t_0(\omega) + m(\omega)S)I_0(\omega), \\ (T_0(\Omega) + M(\Omega)s)(I_0(\Omega) + I_{SHG}) \end{array} \right) \qquad (7)$$

$$\begin{aligned} \mathrm{cov}(I(\omega), I(\Omega)) = &\, t_0(\omega)T_0(\Omega)\mathrm{cov}(I_0(\omega), I_0(\Omega)) \\ &+ t_0(\omega)T_0(\Omega)\mathrm{cov}(I_0(\omega), I_{SHG}) \\ &+ T_0(\Omega)m(\omega)\mathrm{cov}(SI_0(\omega), I_0(\Omega) + I_{SHG}) \\ &+ t_0(\omega)M(\Omega)\mathrm{cov}(I_0(\omega), s(I_0(\Omega) + I_{SHG})) \\ &+ m(\omega)M(\Omega)\mathrm{cov}(SI_0(\omega), s(I_0(\Omega) + I_{SHG})) \end{aligned} \qquad (8)$$

Equation 8 follows from 7 by the bilinearity of covariance. Similar calculations can be done to decompose the correlations between two pixels in the fundamental region or two pixels in the harmonic region. From Eq. 8, we can see that the covariance has contributions from, in order: the intrinsic correlation from the fundamental and XFEL harmonic, the correlation between the fundamental and the SHG signal, the XFEL harmonic and fundamental TA, the fundamental and the harmonic TA, as well as the covariance between the various TA signals and the SHG (comparably small). The dominant contributions from the intrinsic correlations can then be subtracted using the jet-out covariance data multiplied by the linear transmission of the harmonic and

fundamental at that energy, which yields only the intensity-dependent contributions. Neglecting higher-order cross terms in $m$, $M$, and $I_{SHG}$, the difference maps, $D$, are then:

$$D(I(\omega), I(\Omega)) = \mathrm{cov}(I(\omega), I(\Omega)) - t_0(\omega)T_0(\Omega)\mathrm{cov}(I_0(\omega), I_0(\Omega)) \qquad (9)$$

$$\begin{aligned} D(I(\omega), I(\Omega)) \approx &\, t_0(\omega)T_0(\Omega)\mathrm{cov}(I_0(\omega), I_{SHG}) \\ &+ T_0(\Omega)m(\omega)\mathrm{cov}(SI_0(\omega), I_0(\Omega)) \\ &+ t_0(\omega)M(\Omega)\mathrm{cov}(I_0(\omega), sI_0(\Omega)) \end{aligned} \qquad (10)$$

These covariance difference maps with the intrinsic correlations suppressed are the maps which are shown in the main text and analyzed below. Raw jet-in and jet-out covariance maps for this spectral region can be seen in Supplementary Fig. 3 for the 550 eV dataset highlighted in the main text.

Experimentally, the jet-out covariance matrix can then be multiplied element-wise with the outer product of the jet transmission to get the full covariance difference matrix. The same estimated transmission matrix was used for the experimental data and the modeled data (described below) to minimize the errors in fitting the data.

## Modeling the SHG response

While a well-behaved transform-limited pulse in a medium with a uniform $\chi^{(2)}$ response will result in the SHG signal being the convolution of the fundamental spectrum, a poorly behaved pulse results in the SHG response resembling the square of the fundamental profile. These two limiting cases present differently in the covariance maps and binned data, so the response has to be modeled. An empirical definition of $I_{SHG}$ was employed:

$$I_{SHG}(2\omega) = \left|\chi_{eff}(2\omega)\right|^2 \int I_0(\omega - \Delta)I_0(\omega + \Delta)g(\Delta)d\Delta \qquad (11)$$

Where the effective nonlinear susceptibility is only dependent on the output photon frequency and the function $g(\Delta)$ is chosen to interpolate between the two limiting cases of interest for our SHG signal.

For computational efficiency, a boxcar function was used for $g$:

$$g(\Delta) = \left\{ \begin{array}{ll} 1, & |\Delta| \leq \Delta_m \\ 0, & |\Delta| > \Delta_m \end{array} \right\} \qquad (12)$$

Where $\Delta_m$ is the range of frequencies which can interact in a $\chi^{(2)}$ process. The best fit choice for $\Delta_m$ was determined to be about -1.5 eV (7 pixels on the detector) and was used for the covariance model fitting.

Empirically it was found that the nonlinear response had a spectral profile which cut off on the red side of the spectrum (e.g., Fig. 3A). As the spectral response was consistent across all of the potential SHG signal datasets examined, it is likely more reflective of the XFEL source than the nonlinear properties of the water sample. To incorporate the observed spectral drop-off, the nonlinear spectral response was then modeled phenomenologically as a logistic function:

$$\left|\chi_{eff}(\Omega)\right|^2 = \frac{|\chi_0|^2}{1 + \exp(-b(\Omega - \Omega_0))} \qquad (13)$$

Where $|\chi_0|^2$ is the magnitude of the nonlinear response, $b$ is the logistic growth rate (curve steepness) and $\Omega_0$ is the midpoint of the function.

## Data fitting

The TA signal parameters and the derived SHG parameters were determined with a simultaneous fitting of the experimental covariance maps and the binned intensity-dependent features. Broadly, parameters controlling the frequency-dependent values of the linear transmission ($t_0$, $T_0$) and TA features ($m$, $M$) were input and used to

calculate the binned SHG signal based on Eq. 4. The binned SHG response was compared to the calculated SHG response using input-ted values for the SHG spectral response ($\Delta_m$, $\Omega_0$, $b$) to determine the magnitude of the response, $|\chi_0|^2$. A simulated jet-in dataset that was generated from the jet-out dataset and parameters for the various intensity-dependent processes indicated in Eq. 10, which was then used to calculate a model covariance map. The goodness of fit was determined by a composite parameter using the residual of the experimental covariance map to the model map and the residual of the binned SHG signal to the calculated binned SHG response.

As was discussed above and illustrated in Supplementary Fig. 2, estimates for $t_0$, $m$, $T_0$, and $M$ can be obtained from linear fits to the binned data. The spectral response to the fundamental TA $m(\omega)$ was assumed to be broadly spectrally accurate, and only a global scale factor and offset was used for fitting. As the harmonic TA signal is the most uncertain and has the greatest impact on the derived SHG signal, $M(\Omega)$ was modeled as a five-point cubic spline equally spaced in $\Omega$.

The new estimates for $t_0$, $m$, $T_0$, and $M$ were then used to obtain the binned isolated SHG signal per Eq. 4. The binned SHG signal was used to determine the magnitude of $|\chi_0|^2$ by integrating the overall signal in each bin and taking a linear fit between the experimental (Eq. 4) and calculated (Eq. 11) integrated SHG signals with a y-intercept of zero. From here, the transmission and TA for both regions and the full nonlinear spectral response are used to calculate the modeled jet-in data from which the modeled covariance map and the intensity-dependent difference spectra (isolated SHG signals) can be obtained.

The goodness of fit was determined by simultaneously minimizing the squared residual of the experimental and modeled covariance difference maps and the squared residual of the experimental and modeled intensity-dependent isolated SHG signals. The overall composite function that was minimized was:

$$F = \left(1 - R_{cov}^2\right) + \left(1 - R_{bin}^2\right) + \log(r_{bin}^2) \qquad (14)$$

Where $R_{cov}^2$ and $R_{bin}^2$ are weighted coefficients of determination for the covariance map fit and binned SHG signal fits respectively and $r_{bin}^2$ is the squared residual of the binned fits.

The fit results were still found to suffer from systematic errors, particularly in the range of bins considered in the determination of $T_0$ and $|\chi_0|^2$ and the integration bounds considered for determining $|\chi_0|^2$. To attempt to quantify this additional uncertainty, for each dataset 13 different permutations of the bin and integration bounds choices were considered. These systematic errors were found to be comparable or larger to the raw statistical errors in many cases and are included in the error bars of the linear transmission, TA fits, and fits of the nonlinear spectral response, including the nonlinear suscept-ibilities reported in Fig. 3.

The experimental and resulting best fit covariance map, the binned isolated SHG signal, and the quadratic fit for each dataset are presented in Supplementary Fig. 4. Error bars in binned signals are the standard deviation between the three experimental runs. The $R$-squared values used for fitting in Eq. 14 are shown in Supplementary Fig. 5. The covariance map fits were fair to good with $R$-squared values above 0.5. The binned spectra fit and intensity-dependent quadratic fits show similarly fair-to-good fits above the K-edge, but SHG fits were disfavored relative to the no-signal null case for all of the datasets below the K-edge.

The intensity-dependent datasets containing candidate SHG signals were also fit to a power law of the form:

$$y = A|x|^p \qquad (15)$$

Which yielded fits of similar quality to the quadratic fits (Supplementary Fig. 5B, red circles). The best fit power law exponents, $p$, are shown in Supplementary Fig. 5C. The resulting best fit power laws were scattered between 1.5 and 2.5, consistent with quadratic behavior and similar to what is seen in conventional lab based optical SHG measurements.

## Molecular dynamics simulations

Here, we employed the recently developed MB-pol(2023) potential[41] and the LAMMPS[58] MD simulation engine for all our simulations. To date, the MB-pol many-body water potential of Paesani and coworkers[41–43,59–61] is widely considered to be amongst the most accurate, transferable and predictive, able to simultaneously reproduce the bulk and interfacial structure and thermodynamics of the liquid, vapor and solid phases. To start, we generated a unit-cell consisting of 432 water molecules, in a cubic box of length 18.77 Å, packed to minimize the interaction energy by applying the continuous configurational Boltzmann biased Monte Carlo method[62,63]. We then equilibrated the bulk liquid using our standard procedure[64]: after an initial conjugate gradient energy minimization to a root mean square (RMS) force of $10^{-5}$ kcal/(mol·Å), we heated the system from 0 K to the desired temperature (298 K and 277 K) over 100 ps in the constant volume, constant temperature (canonical or NVT) ensemble with a Nose-Hoover chain of 3 thermostats with a relaxation time of 0.25 ps. The system was propagated forward in time using a velocity-Verlet algorithm with an integration time step of 0.5 fs. van der Waals and short range electrostatic interactions were calculated explicitly, with a real space cutoff of 0.9 nm, while the long-range electrostatics were calculated in reciprocal space using the particle–mesh Ewald (PME) method, as implemented in the helPME library[65,66], with a convergence tolerance of $10^{-5}$ kcal/mol. The many-body energies, forces and stresses were calculated in LAMMPS by means of the fix mbx and pair mbx functionality which allows for seamless interfacing with the MBX software[67].

After initial NVT equilibration, we resolved any stresses in the system by means of 1 ns of constant pressure, constant temperature (NPT) simulations using the Andersen barostat (pressure relaxation constant of 1 ps). The equations of motion used are those of Shinoda et al.[68], which combine the hydrostatic equations of Martyna et al.[69] with the strain energy proposed by Parrinello and Rahman[70]. The time integration schemes closely follow the time-reversible measure-preserving Verlet integrators derived by Tuckerman et al.[71] During the last 500 ps of the 1 ns NPT simulation, we calculated the average cell lengths and linearly adjusted the final NPT simulation cell to the averages, over a further 100 ps of dynamics. Finally, we performed production dynamics for an additional 2 ns of NVT dynamics. As a figure of merit, the calculated density of water at ambient conditions (298 K, 1 atm) was 0.997 g/cm$^3$.

The final snapshot of our bulk simulations was used as an initial configuration for our vacuum/water simulations. Here, we inflated the simulation cell in the z-direction by 10 nm and centered the liquid slab within the cell, thus allowing 5 nm of vacuum at each surface. The top and bottom of the simulation box were bounded by a purely repulsive wall, using the fix wall/harmonic functionality in LAMMPS, for stable simulations in the rare event that molecules evaporated from the surface and approached the unit cell z-boundaries. After initial heating, we performed a 2D-NVT simulation for 4 ns. In all cases, the slab was found to be stable over the entire MD simulation. All interface simulation parameters were the same as the bulk simulations, except for the application of the 2D PME method and the 2D slab corrections of Yeh and Berkowitz[72] with a further 2.0 z factor, in order to eliminate spurious interactions between the two surfaces.

To validate our vapor/water simulations, we calculated the interfacial surface tension (IFT), using the rigorous statistical mechanical formulism first proposed by Tolman[73], and later developed more fully by Kirkwood and Buff (KB)[74]:

$$\gamma = \frac{1}{2}\int\left[p_\perp(z) - \frac{1}{2}\left\{p_{\|a}(z) + p_{\|b}(z)\right\}\right]dz \qquad (16)$$

where $p_\perp$ is the component of the stress tensor perpendicular to the surface (z-axis in our coordinate system), while $p_{\|a}$ and $p_{\|b}$ are the parallel (x- and y-axis) components. The general idea behind this KB method is that in the bulk liquid, the parallel and perpendicular stress components are equal and cancel, while at or near the interface, $2p_\perp > p_\|$. Thus Eq. 16 allows the IFT to be obtained from MD calculations by a simple integration of the components of the pressure. The calculated IFT using this approach was $69.5 \pm 1.3$ mJ/m$^2$ at 298 K and $72.6 \pm 2.5$ mJ/m$^2$ at 277 K. While the calculated IFT is slightly under-predicted compared to experiments (72.0 and 75.1 mJ/m$^2$ respectively), the calculated slope of the IFT with decreasing temperature of 1.045 mJ/m$^2$/K is in excellent agreement with the value calculated from the experimental steam tables[75] of 1.042 mJ/m$^2$/K, suggesting that the surface entropy, and by extension the interfacial structure, of MB-pol(2023) is representative of the experimental reality.

## Mass density distribution and interface definition

We employ a rigorous definition of the water interface based on the instantaneous interface definition advanced by Willard and Chandler[44]. Here the interface is defined by a time-dependent density field $\rho(\mathbf{r},t)$, constructed from Gaussian functions located at the water molecules' center of mass $\mathbf{r}_i(t)$:

$$\rho(r,t) = \sum_i^{N/3} \left(2\pi\xi^2\right)^{-\frac{3}{2}} \frac{1}{\sqrt{e}} \left(\frac{r - r_i(t)}{\xi}\right)^2 \quad (17)$$

where $\xi = 0.24$ nm (~ the diameter of a water molecule) is the Gaussian width, representing a system-dependent coarse-graining length.

From Eq. 17 we can define the instantaneous interface at a given time $t$ as the 2D manifold $\mathbf{s}(t) = r$, for which the density field is constant. In the current work, we take this constant to be half the bulk density, i.e., the Gibbs dividing surface. Once defined, we apply a smoothing function, by means of a cubic interpolation scheme, to ensure a continuous instantaneous interface definition. Then the position of each water molecule was taken as the closest contact point to each of the two instantaneous interfaces, discretized along the z-axis in 0.1 Å bins. An example of the resulting profiles is given in Supplementary Fig. 6B. There we find that while the planar interface shows the usual hyperbolic tangent dependence with distance, there is instead significant oscillations in the density according to the instantaneous interface definition. Moreover, we are able to unambiguously define the 1st interface layer, as the region extending 0.67 nm into the liquid from the point of vanishing density.

## Hydrogen bond definition and analysis

We employed a parameter free, radial-angle joint distribution function $g(R,\beta)$ (shown schematically in Supplementary Fig. 6C), which is the normalized probability of observing an acceptor-donor pair in a volume element between $R$ and $R + dR$, and an acceptor-donor-hydrogen angle between $\beta$ and $d\beta$:

$$g(R,\beta) = \Pr(R,\beta)/Q$$
$$Q = \frac{N(N-1)}{V} 2\pi \sin\beta d\beta R^2 dR \quad (18)$$

where $Q$ is the expected probability if the system was noninteracting[76], $N$ is the number of oxygen donor/acceptor atoms, and $V$ is the volume. We then calculate the potential of mean force (PMF) $W(R,\beta)$ as:

$$W(R,\beta) = -kT\ln(g(R,\beta)) \quad (19)$$

Notably, for consideration of the vapor/water interface, we determined the $<V_{interface}>$ and $<N_{interface}>$ from the mass-density profile of the instantaneous interface in the previous section.

We defined the H-bonded state as an equipotential region of the 2D PMF surface that passes through the saddle point, and encircles the minimum, while all other combinations are considered to be broken H-bonds, or in the extreme case, free OH-bonds (i.e., water molecules not participating in any H-bonds). Practically, this amounts to first constructing $g(R,\beta)$ and numerically determining $B(R,\beta,W_{cut})$: the bounding region encircling a minima with energies within the cutoff $W_{cut}$, numerically. Specifically, $W_{cut}$ was determined self-consistently as the saddle point where the overlap between two or more minima $B(R, \beta, W_{cut})$ is exactly 1. In the current case of MB-pol(2023), we determined that for the bulk liquid at 298 K and 1 atm, $W_{cut} = 1.22$ kT, which we show graphically as the shaded region in Supplementary Fig. 6C. The H-bonding statistics were then obtained by reanalyzing the MD trajectory and selecting only the water H-bonds within $B(R,\beta,W_{cut} = 1.22$ kT).

For theoretical estimates of the average number of H-bonds per water molecule $<n_{HB}>$, we decomposed the H-bond configurations for the bulk and interface layer into the 9 Acceptor(A)/Donor(D) configurations $A_xD_y$, where $x, y \in [0, 1, 2]$. As a figure of merit, we calculated an average number of H-bonds of $<n_{HB} \geq 3.55 \pm 0.12$ for our bulk simulations at 298 K, which reduces to $<n_{HB} \geq 2.92 \pm 0.45$ in the 1st interfacial layer, defined based on our instantaneous interface calculations described above. The percentage of the various H-bonding configurations in the bulk and at the interface at 298 K and 277 K are given in Supplementary Table 1.

## X-ray absorption spectra simulation

Oxygen K-edge spectra were calculated by exciting each oxygen atom in each snapshot individually using constrained-occupancy DFT calculations employing the PBE GGA functional[77]. Plane–wave pseudo-potential calculations using ultrasoft pseudopotentials[78] were performed using the PWSCF code within the Quantum-ESPRESSO package[79]. We used a kinetic energy cut-off for electronic wave functions of 25 Ry and a density cut-off of 200 Ry. The core-excited Kohn–Sham eigenspectrum was generated using the XCH approach[80]. Based on a numerically converged self-consistent charge density, we generated the unoccupied states for our XAS calculations non-self-consistently, sufficiently sampling the first Brillouin zone with a $2 \times 2 \times 2$ uniform k-point grid, employing an efficient implementation of the Shirley interpolation scheme[81] generalized to handle ultrasoft pseudopotentials[82]. Matrix elements were evaluated within the PAW frozen-core approximation[83]. Core-excited ultrasoft pseudopotentials and corresponding atomic orbitals were generated with the Vanderbilt code[78]. Each computed transition was convolved with a 0.2 eV Gaussian function to produce continuous spectra.

The use of pseudopotentials in our calculations means that we lose the absolute reference state from which we can base our linear excitation energies, as would be present in an all-electron calculation. Thus, each spectra needed to be properly calibrated for unambiguous comparisons, especially in systems with varying number of molecules, total charge, or different cell sizes. As in our previous work[84], we here apply two independent calibration schemes. First, we apply to the formation energy formulism of Prendergast and coworkers to self-consistently align our spectra to a common reference[85]. Previous work has shown that this approach is accurate in predicting the main peak position of various lithium compounds to within 0.1 eV of the experiments[86]. We then applied an absolute energy shift, by matching the simulated Oxygen K-edge white-line of $O_2$ to the experiments, which amounted to a shift of +535.5 eV applied to all our simulated spectra.

## Nonlinear SXSHG simulations

To demonstrate the second harmonic susceptibility of the water interface, first principles simulations of the SHG were performed. X-ray simulations become prohibitively expensive as the system size

increases, which limits the absolute size of the model systems. Here, we used simulation cells in a slab geometry with dimensions $10 \times 10 \times 30$ Å and the total number of atoms ranging from 90–110 depending on the surface configuration. 20 Å of vacuum was included to represent the water/vacuum interface and to avoid periodic image interactions. These smaller unit cells were then propagated forward in time using our approach outlined above, with harmonic restraints applied to the internal structure and center of mass position of the particular H-bond configuration (i.e., the $A_xD_y$ species and its first solvation shell). This allowed for the equilibration of the water structure around the $A_xD_y$ configuration, which we found lead to rapidly convergent electronic-structure calculations. Next, we performed excited state calculations using velocity-gauge real-time time-dependent density functional theory (RT-TDDFT) as implemented in a modified version of the SIESTA package[40], with a double-$\zeta$ with polarization atomic natural orbital basis set used to model electronic structures. One of the advantages of performing RT-TDDFT in the velocity gauge is that the method provides a full multipole expansion via the applied external vector potential and the resultant time-dependent current density. In this sense, all contributions, dipole, quadrupole, etc., are included, albeit at a nontrivial computational cost, meaning the method is also sensitive to bulk-active contributions to the signal.

Soft X-ray transitions were obtained by including the oxygen 1s orbital such that the solutions of Kohn-Sham DFT contained core electrons. The system was sampled at the gamma point, with a plane wave energy cutoff of 400 Ry. As SIESTA uses the real-space atomic natural orbital basis rather than plane waves, this cutoff corresponds to the size of the real-space grid that the relevant objects are represented on in analogy to how dense such a grid would have to be for a corresponding plane wave basis.

The second-order susceptibility of the water surface was obtained by propagating an intense laser pulse through the system and calculating the resulting current. A perturbative pulse of 521 eV was used with the timestep of 0.0055 a.u., corresponding to -0.2661 attoseconds. The resulting non-equilibrium electron density was propagated under the full pulse duration of 1 fs to obtain the second harmonic response of the system. The SHG response of the water surface was extracted following the semi-slab approach established in our previous works[20–24,87].

To extract the magnitude of the second harmonic response, we performed SHG calculations with pulse intensities from $1.0 \times 10^{12}$ to $7.5 \times 10^{17}$ W/cm$^2$. The current density of the system due to perturbation can be described as a Taylor series expansion of field strength:

$$j = \chi^{(0)} + \chi^{(1)}E(\omega) + \chi^{(2)}E(\omega)^2 + \cdots \tag{20}$$

The resulting response at $2\omega$ was obtained from numerical integration and fit to a quadratic in the intensity. It has been shown that the total $\chi^{(2)}$ response can be expressed as an outer sum of the initially occupied states[88], which are here the individual O 1s orbitals. This allows the SXSHG response to be calculated for each water molecule, which allows the contributions to the full SXSHG spectrum from different H-bond configurations to be determined.

We found that the X-ray SHG is dominated by the topmost water molecules at the interface, with a signal that is more than an order of magnitude larger than the corresponding simulated spectra in the bulk (3D geometry), even with the limitation of the small number of water molecules included in the simulation. A representative calculated current, across the fundamental and harmonic for the 3D bulk and 2D slab geometries are shown in Supplementary Fig. 7A. A plot of the $2\omega$ response as a function of field strength is demonstrated in Supplementary Fig. 7B. A comparison of the full simulated SXSHG spectrum to the simulated XAS is shown in Fig. Supplementary Fig. 7C, showing the experimental shift is replicated. The simulated SXSHG spectra for

each major H-bond species was shown in Fig. 4C. The remaining minor species (surface prevalence of <3% in Supplementary Table 1) are shown in Supplementary Fig. 7D.

## Quantum chemistry calculations

To try and develop an understanding of the possible states being accessed with second harmonic excitation, first principles spin-unrestricted linear response TD-DFT calculations were performed, using the B3LYP functional[89] within the Q-Chem 6.2 electronic structure package[90]. These simulations used a reduced excitation space[91] of just the oxygen 1s electrons. Basis set polarization at the 5-$\zeta$ level within the Dunning correlation consistent basis sets[92] (i.e., cc-pV5Z) was required to have sufficient density of states at and above twice the O K-edge to determine the possible second harmonic excited states. Initial TD-DFT simulations were performed on only a single water molecule to calculate all possible TD-DFT excited states. Within this methodology, the lowest lying O 1s excited state with a non-negligible oscillator strength (i.e., the K-edge) was found to be around 518.9 eV. Thus, the excited states considered as potentially accessible at the second harmonic began at -1040 eV and are listed in Supplementary Table 2. To visualize these states, their first pair of natural transition orbitals were determined, along with a corresponding cube file. Although the calculations are spin-unrestricted, the spin-up and spin-down natural transition orbitals were identical, and only the spin-up versions are presented here.

## Data availability

The experimental data as shown in the main text and Supplementary figures, the TDDFT input and data files, and molecular dynamics simulation input and data files are available on Zenodo[93].

## Code availability

Custom code used for TDDFT and molecular dynamics simulations can be found in the open-access repository on Zenodo[93].

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

## Acknowledgements

D.J.H. acknowledges helpful discussions with L. Young, T.J.A. Wolf, and J.P. Cryan. T.A.P. acknowledges fruitful discussions from V.A.P. and L.J.P. Use of the Linac Coherent Light Source (LCLS), SLAC National Accelerator Laboratory, is supported by the U.S. Department of Energy, Office of Science, Office of Basic Energy Sciences under Contract No. DE-AC02-76SF00515. This research used resources of the National Energy Research Scientific Computing Center, a DOE Office of Science User Facility supported by the Office of Science of the US DOE, under contract no. DE-AC02-05CH11231, using NERSC award BES-ERCAP0033755. This work also used the Expanse supercomputer at the San Diego Supercomputing Center through allocations DMR190106 and PHY210131 from the Advanced Cyberinfrastructure Coordination Ecosystem: Services & Support (ACCESS) program, which is supported by NSF Grant No. 2138259, No. 2138286, No. 2138307, No. 2137603, and No. 2138296. D.J.H. and J.D.K. were supported by the Department of Energy, Laboratory Directed Research and Development program at SLAC National Accelerator Laboratory, under contract DE-AC02-76SF00515. M.W.Z. acknowledges funding by the Department of Energy (DE-SC0024123). M.W.Z., T.A.P., J.D.K., and R.J.S. acknowledges funding from the UC Office of the President within the Multicampus Research Programs and Initiatives (Grant No. M21PL3263). W.S.D. was supported by the U.S. Department of Energy, Office of Science, Basic Energy Sciences, Chemical Sciences, Geosciences, and Biosciences Division under Contract DE-AC02-05CH11231, FWP number FP00014920. S.W.D., D.S., K.V.L., and C.P.S. were supported by the U.S. Department of Energy, Office of Science, Basic Energy Sciences, Chemical Sciences, Geosciences, and Biosciences Division under Contract DE-AC02-05CH11231. F.B., S.W.D., and R.J.S. were supported by the Department of Energy, Office of Basic Energy Sciences, through the Chemical Sciences Division at the Lawrence Berkeley National Laboratory under contract #CH403503. S.J. and A.D. were supported by the US DOE BES grant no. DE-SC0023503. S.J. acknowledges the financial assistance award 70NANB23H111 from U.S. Department of Commerce, National Institute of Standards and Technology. B.R.N. acknowledges funding from the National Science Foundation Graduate Research Fellowship Program (Grant No. DGE 1752814). J.A.S. acknowledges support from the Arnold O. Beckman Postdoctoral Fellowship Program. E.J.R. acknowledges support from CALSOLV/RESOLV-Bochum and the UC Office of the President within the Multicampus Research Programs and Initiatives. T.A.P. acknowledges support from the Sloan Research Foundation.

## Author contributions

C.P.S., W.S.D., R.J.S., M.W.Z., and J.D.K. originated the project concept. D.J.H., S.W.D., D.G., J.A.S., B.R.N., F.B., E.J.R., K.K., C.Y.H., G.L.D., C.P.S., M.W.Z., and J.D.K. executed the experiment and collected data. J.D., D.C., N.S., and A.M. coordinated attosecond pulse delivery. D.J.H. and S.W.D. provided sample characterization. D.J.H., S.W.D., D.G., and J.D.K. analyzed experimental data. S.J., D.S., A.D., K.V.L., and T.A.P. provided theoretical modeling. D.J.H., K.V.L., T.A.P., C.P.S., M.W.Z., and J.D.K. wrote the manuscript with contributions from all authors.

## Competing interests

The authors declare no competing interests.
