## [Transparent Peer Review file · Nature Communications]

Surface structure of water from soft X-ray second harmonic generation

Corresponding Author: Dr David Hoffman

Version 0:

Reviewer comments:

Reviewer #1

(Remarks to the Author)

The manuscript by Hoffman et al. presents a combined experimental and theoretical investigation of soft X-ray second harmonic generation (SXSHG) at the air/water interface. The authors report state-of-the-art SXSHG measurements and demonstrate that this spectroscopy can selectively probe interfacial water molecules with specific hydrogen-bonding motifs. The successful application of this technique to a liquid interface, as demonstrated here, opens exciting possibilities for studying interfacial chemistry in aqueous systems.

The structure and dynamics of water at interfaces remain an important and actively researched topic. I find significant merit in this work; however, several key points must be addressed before I can recommend it for publication:

1. The manuscript mentions that the transmission setup used is simpler. However, the authors do not justify why a reflection setup—such as the one used by some of the co-authors in *Phys. Rev. Lett.* 120, 023901 (2018)—was not considered. Please elaborate on the rationale behind the choice of geometry.

2. Related to the previous question, given that the experiment employs a flat microjet, have the authors considered possible interference between SHG signals generated at the front and back interfaces? If such interference is negligible, please explain why. Otherwise, can the signals be disentangled or their contribution estimated?

3. The authors attribute the transient absorption signal to intrapulse ionization of the water sample. This signal appears to have a comparable magnitude to the SXSHG response and could thus significantly affect the interpretation of the results. On page 6, the authors suggest the signal is "likely to be flat" but provide no further support. However, Extended Data Fig. 2 (page 46) shows spectral features that contradict this assumption, which the authors attributed to contamination with the SXSHG signal. Given its crucial characterization, the authors should rigorously evaluate how these assumptions employed during the background subtraction might affect the SXSHG spectra shown in Figure 3C.

4. I am not fully convinced by the explanation for the weak SXSHG response of tetrahedral double-donor-double-acceptor species. The authors attribute it to a "high degree of symmetry," but similar species are known to contribute strongly to sum-frequency generation (SFG), which shares similar selection rules regarding molecular symmetry. Please clarify this point and reconcile it with prior SFG studies.

5. Since theoretical modeling plays a central role in interpreting the experimental results, the authors should provide a more detailed justification of their computational choices, and potentially additional convergence and robustness tests, — particularly regarding the plane-wave energy cutoff and the exchange-correlation functional.

6. It is unclear how the contributions from water molecules with different hydrogen-bonding motifs are disentangled in the simulations. Since the current density induced by the laser is a collective response of the entire system (Eq. 20), what criteria or computational strategy were used to decompose the signal as shown in Figure 4C? This figure is critical to the manuscript's conclusions and requires a more thorough explanation.

(Remarks on code availability)

Reviewer #2

(Remarks to the Author)

In this work, Hoffman and co-workers investigate the H-bonding structure of interfacial water at the vapor/liquid interface by developing soft X-ray second harmonic generation (SXSHG). The key technical advancements of the study are as follows:

A. The use of the X-ray Laser-Enhanced Attosecond Pulses (XLEAP) technique to generate high-peak-power, sub-femtosecond laser pulses.

B. The implementation of a flat, thin liquid sheet microjet to detect both the fundamental and harmonic signals in a transmission geometry.

The main claim of the study is that the SXSHG signal measured at the vapor/water interface is blue-shifted compared to the bulk water X-ray absorption spectrum (XAS), implying that singly hydrogen-bonded water molecules are more prevalent at the interface.

While I find this technique promising as a new approach for probing interfacial water structure, and the observation itself is interesting, I have the following concerns that should be addressed to further support the authors' claims and broaden the paper's appeal:

1. Background and Motivation for a General Audience:

Given that Nature Communications targets a broad readership, it would be constructive for the authors to provide more background and motivation regarding both the scientific question and the technical approach. For instance, what is the current status of soft X-ray SHG? What specific advantages does this technique offer over conventional SHG? A brief explanation of the oxygen K-edge and its spectroscopic relevance would also be helpful for non-specialists.

2. Interpretation and Broader Impact:

The manuscript lacks a deeper interpretation of the results and their implications. The conclusion that interfacial water has a different hydrogen-bonding structure than bulk water seems rather expected. As the authors state in the introduction, "Despite the pervasiveness of aqueous interfaces and the many tools developed to study them, there are still substantial open questions about their electronic and molecular structure and how they connect to important macroscopic phenomena." It would strengthen the paper if the authors explicitly discussed how their results help address these open questions or provide new insights into the link between interfacial molecular structure and macroscopic properties.

3. Spectral Quality and Data Robustness:

A central result of the study is the SXSHG spectrum shown in Fig. 3C. However, the large error bars and limited number of data points raise concerns about the reliability of the result. As it stands, the spectral quality does not support even semi-quantitative comparisons or analysis. It would be helpful if the authors could provide additional technical and sample replicates to enhance the validity of their findings. Furthermore, a forward-looking discussion on how the spectral quality and signal-to-noise ratio might be improved in future measurements would be valuable. As the field of interfacial water is already subject to considerable debate, limited data quality may risk introducing further inconsistencies rather than resolving them.

(Remarks on code availability)

I am not an expert in modeling and therefore defer to the evaluation and judgment of other reviewers on this aspect of the work.

Reviewer #3

(Remarks to the Author)

The manuscript innovatively reports soft X-ray second harmonic generation (SXSHG) from the water/vapor interface to elucidate interfacial electronic structure. Key findings reveal distinct SXSHG spectra compared to bulk X-ray absorption, highlighting broken hydrogen-bonding networks at the surface. Combining theoretical calculations, the results indicate that the signal is highly sensitive to single-acceptor H-bond configurations, abundant at the interface. This study opens up new possibilities for studying molecular properties at interfaces using nonlinear X-ray spectroscopy, with significant implications for understanding interfacial chemistry and improving theoretical models of liquid interfaces. Given recent advances in flat liquid sheet microjets and XFEL technology, the study is very timely and significant for advancing surface science.

While the results carry scientific significance and I can recommend the work for publication in principle, a several critical aspects of the manuscript require improvement and clarification to reinforce their conclusions and elevate the manuscript quality.

1. Surface sensitivity

The SXSHG in transmission geometry is employed in this study. As we know that the transmitted sum frequency vibrational spectroscopy and optically SHG is not necessarily surface sensitive due to the bulk quadrupole contribution and longer interaction length. The authors should evaluate the bulk quadrupole contribution in the transmitted SXSHG measurement, which is critical for surface sensitivity on water/vapor of the technique.

2. Physical Mechanism Underpinning H-Bond Sensitivity

In the manuscript, the simulation results showed that SXSHG spectra is particularly sensitive to the number of acceptor H-bonds, in contrast to linear XAS measurements, which is sensitive to the degree of donor H-bonds. A deeper physical explanation is needed.

3. Fluence dependent measurement

The fluence dependence measurement data were collected in a range of near ~ 0 uJ to 100s uJ as shown in figure 3B and the authors mentioned that "the intrinsic shot-to-shot fluctuations of the XFEL pulse intensity were used to generate the fluence dependence for identifying nonlinear signals". If I did not misunderstand, the pulse energy of every shot is completely random and uncontrollable and fluctuate in a large range shot-to shot? Then, I have a technical puzzle that why

the pulse energy of XEFL fluctuate so large? Given that the large energy fluctuation, how will its spectra profile change? This is critical for the spectral resolution.

One more small suggestion: it would be better to put a sentence stating GMD energy measurements are performed in situ and do not influence the shot energy. This will be friendly for the readers who are not the X-ray field.

Minor comments:

1. There are dual red ribbons in figure 2d. What's the origination of the other one?
2. In the text, the authors use the descriptions of fundamental fluence, intensity and pulse energy in the fluence dependence section. I believe they express the same thing. In figure 3b, it's better to use pulse energy and uJ for x-axis, so it can be directly related to figure 3a. Moreover, the data in figure 3b seems not be the same with what showed in extended figure 4 at 550eV (where the error-bars are obviously not the same.) Please check the data and specify the difference.
3. In page 8, line 186, the description of "intense main peak at 542.5 eV" is not in line with the figure 4c for A1D0, where the peak is at 544 eV. The authors should check the consistency.
4. What's the chamber pressure for this SXSHG measurement? Please specify this parameter in the methods.

(Remarks on code availability)

Version 1:

Reviewer comments:

Reviewer #1

(Remarks to the Author)

The authors have addressed the comments by the reviewers satisfactorily, I recommend publication of the manuscript.

(Remarks on code availability)

Reviewer #2

(Remarks to the Author)

I find the revised manuscript to be significantly improved, with the authors addressing all questions. The technique presented shows strong potential for future investigations of interfacial systems, and the current work serves as an important proof of principle. That said, improved spectral quality and more detailed molecular-level interpretations will be valuable in future studies.

I have no further questions or requests, and I recommend the publication of this manuscript.

(Remarks on code availability)

Reviewer #3

(Remarks to the Author)

The authors have addressed all my concerns, and I think this manuscript can be accepted as it is.

(Remarks on code availability)

REVIEWER COMMENTS

Reviewer #1 (Remarks to the Author):

The manuscript by Hoffman et al. presents a combined experimental and theoretical investigation of soft X-ray second harmonic generation (SXSHG) at the air/water interface. The authors report state-of-the-art SXSHG measurements and demonstrate that this spectroscopy can selectively probe interfacial water molecules with specific hydrogen-bonding motifs. The successful application of this technique to a liquid interface, as demonstrated here, opens exciting possibilities for studying interfacial chemistry in aqueous systems.

The structure and dynamics of water at interfaces remain an important and actively researched topic. I find significant merit in this work; however, several key points must be addressed before I can recommend it for publication:

1. The manuscript mentions that the transmission setup used is simpler. However, the authors do not justify why a reflection setup—such as the one used by some of the co-authors in *Phys. Rev. Lett.* 120, 023901 (2018)—was not considered. Please elaborate on the rationale behind the choice of geometry.

The geometry is simpler because the X-ray spectrometer can be aligned without using the inherently weak X-ray reflection from the liquid target. This allows for full characterization of the beamline with the same spectrometer used for the measurement. It also does not depend on using the liquid jet as an optic, which makes the alignment more stable over time and less sensitive to any sample fluctuations.

Additionally, this geometry makes it much simpler to tune the angle of incidence of the x-ray beam to the target, as high AOIs (like the 70 deg used here) are usually beneficial for producing greater surface SHG signals. The chemRIXS instrument used for this measurement is limited to reflections spectrometers mounted at 90 or 45 deg, corresponding to a 45 deg or 22.5 deg AOI. The transmission geometry also allowed for the transmission XAS to be collected simultaneously with the SXSHG, removing uncertainty in X-ray energy calibration.

Edits:

pg 4: "...in transmission geometry, which greatly simplifies the experimental design by allowing for alignment of the spectrometer without the liquid target, minimizing the impacts of possible jet fluctuations, and enabling for the simultaneous measurement of the XAS and SXSHG from the same sample."

2. Related to the previous question, given that the experiment employs a flat microjet, have the authors considered possible interference between SHG signals generated at the front and back interfaces? If such interference is negligible, please explain why. Otherwise, can the signals be disentangled or their contribution estimated?

Interference between the two interfaces of the sheet have been seen to occur using optical SHG, which we examined in a recently submitted manuscript. For the soft X-ray SHG examined here, the fundamental is strongly absorbed by the liquid sheet, even with the sub-micron thickness. As can be seen in Fig. 3, all of the observed SXSHG occurs where the absorption of the water sheet is >1 OD. Since the SXSHG is quadratic in the intensity of the X-rays, this means that the SHG produced by the surface facing the X-ray beam is $>100x$ stronger than that produced by the back face.

Edit:

p. 17: “As the fundamental intensity is strongly attenuated by the sheet ($OD > 1$ at all energies we find SHG signal), there is a greater than 100-fold reduction in the SHG signal generated at the back interface. Therefore any SHG produced by the back interface has a negligible contribution on the measured signals.”

3 .The authors attribute the transient absorption signal to intrapulse ionization of the water sample. This signal appears to have a comparable magnitude to the SXSHG response and could thus significantly affect the interpretation of the results. On page 6, the authors suggest the signal is "likely to be flat" but provide no further support. However, Extended Data Fig. 2 (page 46) shows spectral features that contradict this assumption, which the authors attributed to contamination with the SXSHG signal. Given its crucial characterization, the authors should rigorously evaluate how these assumptions employed during the background subtraction might affect the SXSHG spectra shown in Figure 3C.

The reviewer was right to point out the oversimplification in the text. By “likely to be flat” we meant that the TA at 1000-1100 eV is nonresonant, so it should not have sharp features.

However, the XFEL fundamental and harmonic have a complex temporal and spatial profile relative to each other, so we expect an instrument response that is more complicated. Because of this extra complexity, the background subtraction was vetted by simultaneous fitting of the covariance map as described in the Materials and Methods, as the off-diagonal components reflect (non-resonant) TA processes and the on-diagonal components reflect the SHG process. The statistical and estimated systematic errors from this are represented in the error bars in Fig. 3C.

We have clarified “likely to be flat” in the text and thank the referee for pointing out this inconsistency.

Edits:

p. 7: “as the TA signal must be nonresonant across the harmonic region and the absorption across the fundamental is similarly flat above the oxygen K-edge, although the measured behavior of the TA depends on the fundamental and harmonic pulse profiles.”

p. 8: “The y-error bars in Fig. 3C capture the statistical error as well as estimated systematic error in modeling the TA background process, detailed in materials and methods.”

4. I am not fully convinced by the explanation for the weak SXSHG response of tetrahedral

double-donor-double-acceptor species. The authors attribute it to a "high degree of symmetry," but similar species are known to contribute strongly to sum-frequency generation (SFG), which shares similar selection rules regarding molecular symmetry. Please clarify this point and reconcile it with prior SFG studies.

The referee provides an interesting point which we likely cannot provide a full answer for in this work. One notable difference in the vSFG activity of the A₂D₂ species is that the OH mode is both strongly IR and Raman active, and it is only suppressed in the final vSFG spectrum because of phase cancellation between species oriented in different directions in the surface layers.

By contrast, in the water XAS spectrum, simulations have shown that fully-hydrogen bonded species have reduced brightness, because the valence molecular orbitals in the fully H-bonded species gain more s-character, reducing the probability of transition from the O 1s state. This would reduce the activity of the species in the SXSHG spectra as well without relying on the same kind of phase cancellations. Finally, the SXSHG spectra are purely electronic, so that any expected contributions due to vibronic terms (as in vSFG) will not be present, as the associated timescales are several orders of magnitude different.

Edit:

p. 12: "The weak SXSHG activity of the A₂D₂ species provides an interesting contrast to the vibrational SFG data, as the OH stretch of individual species provide a strong response in the SFG which is minimized in the final spectrum due to phase cancellations due to the opposing orientations of molecules near the surface^{26,51-53}. Here the per-species response is reduced, possibly due to the valence molecular orbitals gaining more s-character when tetrahedrally bonded, as has been seen in simulations of the water XAS spectrum⁵⁴. This may reduce either the one or two-photon cross-section in the symmetric A₂D₂ species."

5. Since theoretical modeling plays a central role in interpreting the experimental results, the authors should provide a more detailed justification of their computational choices, and potentially additional convergence and robustness tests, —particularly regarding the plane-wave energy cutoff and the exchange-correlation functional.

We thank the reviewer for this comment, and the primary justification for the simulation parameters has come from the team's previous experience in modeling these systems, in particular this work on a buried boron interface (<https://journals.aps.org/prl/abstract/10.1103/PhysRevLett.127.096801>), which delved deeply into the parameterization of these kinds of calculations, and this work on Ti XUV-SHG (<https://www.science.org/doi/10.1126/sciadv.abe2265>), which compared the results of this method with the other established linear response approach.

In the underlying SIESTA DFT package being used for the RT-TDDFT simulations, the basis set is atomic natural orbitals (ANOs) rather than plane-waves, and those are expanded as a Bloch-LCAO basis to form the real-space Kohn-Sham one-particle wavefunctions that comprise the time propagated wavepacket. Here, we used a double- ζ with polarization basis set, which is the largest basis that we can use in the version of SIESTA our RT-TDDFT is built on, and it has been shown to produce reasonable results in the team's previous efforts and the initial SIESTA

RT-TDDFT implementation publication

(<https://www.sciencedirect.com/science/article/abs/pii/S0010465518300262>).

The “plane-wave cutoff” applies not to anything relating to the basis, rather it is the ‘MeshCutoff’ keyword in SIESTA which governs the size of the real-space grid that objects such as the density are represented on, and it is supposed to be an analogy to the size of the real-space mesh that would be determined in a plane-wave code. 400 Ry is quite dense, and again it has been previously found to adequately represent the system.

Turning to the functional, the primary states of interest in the Kohn-Sham wavepacket are the O 1s states. These states are buried so deep and atomic-like that any functional does a reasonable job of reproducing them. Since the RT simulation is only propagating the one-particle Kohn-Sham wavefunctions in time which are expanded over the underlying time-independent ANOs, and we’ve found that many of the other issues revolving around functional choice in DFT largely don’t appear and that the LDA and PBE functionals work equally well in reproducing these SXSHG spectra, so as long as the underlying real-space grid (and ANOs) are sufficiently dense (and diffuse) enough to capture how the core states perturb. A manuscript detailing these findings is currently being finalized.

Turning to the single molecule and cluster calculations: these were done with the B3LYP functional, as the simulations were linear response calculations that depended on the quality of the core, valence, and highly excited one-particle Kohn-Sham states and the use of a hybrid functional wasn’t prohibitively expensive due to the smaller system size.

A note about the plane-wave cutoff has been added to the methods section.

Edits:

p. 29: “we performed excited state calculations using velocity-gauge real-time time-dependent density functional theory (RT-TDDFT) as implemented in a modified version of the SIESTA package⁴⁰, with a double- ζ with polarization atomic natural orbital basis set used to model electronic structure.”

p. 30: “As SIESTA uses the real-space atomic natural orbital basis rather than plane waves, this cutoff corresponds to the size of the real-space grid that the relevant objects are represented on in analogy to how dense such a grid would have to be for a corresponding plane wave basis.”

6 It is unclear how the contributions from water molecules with different hydrogen-bonding motifs are disentangled in the simulations. Since the current density induced by the laser is a collective response of the entire system (Eq. 20), what criteria or computational strategy were used to decompose the signal as shown in Figure 4C? This figure is critical to the manuscript’s conclusions and requires a more thorough explanation.

The RT-TDDFT calculations here determine the response for only a single O atom’s excitation in a snapshot per simulation. By doing several RT simulations on the same snapshot at the same energy but at different field strengths, one can then fit $\chi^{(2)}$ for that O’s 1s electrons via expression

20 in the supplemental, ie. $j(2\omega) \propto \chi^{(2)} E(\omega)^2$. Sharma and Ambrose-Draxl (<https://iopscience.iop.org/article/10.1238/Physica.Topical.109a00128>) have shown that through perturbation theory the various components of, and thus the total, SHG $\chi^{(2)}$ can be expressed as summations where there is an outer sum over initially occupied states of interest. In essence, we have taken a similar approach by calculating the $\chi^{(2)}$ from only a single water molecule per snapshot and then summing each of those individual $\chi^{(2)}$'s to get the total $\chi^{(2)}$. In each RT-TDDFT simulation, the particular O's 1s electrons are allowed to interact with all other valence and conduction states, so the other two sums within the expressions for $\chi^{(2)}$ are complete within the limitations of the smaller box sizes needed for these calculations. Therefore, the approach we have taken is a natural decomposition of the summation expressions for SHG $\chi^{(2)}$ although just done in a slightly different approach, and the limitations on available valence and conduction states by the smaller simulation box size doesn't seem to have a large effect on the results. In another forthcoming theory work, we show the equivalence between summing over individual molecules in the RT-TDDFT formalism and the more established linear response approach.

Edits:

p. 9: "The full simulated signal is generated on a per-atom basis, allowing for the contributions from particular molecules to be isolated."

p. 30: "It has been shown that the total $\chi^{(2)}$ response can be expressed as an outer sum of the initially occupied states⁸⁸, which are here the individual O 1s orbitals. This allows the SXSHG response to be calculated for each water molecule, which allows the contributions to the full SXSHG spectrum from different H-bond configurations to be determined."

Reviewer #2 (Remarks to the Author):

In this work, Hoffman and co-workers investigate the H-bonding structure of interfacial water at the vapor/liquid interface by developing soft X-ray second harmonic generation (SXSHG). The key technical advancements of the study are as follows:

A. The use of the X-ray Laser-Enhanced Attosecond Pulses (XLEAP) technique to generate high-peak-power, sub-femtosecond laser pulses.

B. The implementation of a flat, thin liquid sheet microjet to detect both the fundamental and harmonic signals in a transmission geometry.

The main claim of the study is that the SXSHG signal measured at the vapor/water interface is blue-shifted compared to the bulk water X-ray absorption spectrum (XAS), implying that singly hydrogen-bonded water molecules are more prevalent at the interface. While I find this technique promising as a new approach for probing interfacial water structure, and the observation itself is interesting, I have the following concerns that should be addressed to further support the authors' claims and broaden the paper's appeal:

1. Background and Motivation for a General Audience:

Given that Nature Communications targets a broad readership, it would be constructive for the authors to provide more background and motivation regarding both the scientific question and the technical approach. For instance, what is the current status of soft X-ray SHG? What specific advantages does this technique offer over conventional SHG? A brief

explanation of the oxygen K-edge and its spectroscopic relevance would also be helpful for non-specialists.

We thank the referee for this suggestion. We have added a brief overview of the current state of SXSHG literature. We have also increased our explanation of the advantages and drawbacks of the technique as compared to other forms of nonlinear optics. In brief, SXSHG is a relatively recently developed field and has advantages of elemental specificity and being highly sensitive to the chemical environment. The primary downside at present is the measurements can only be performed at large scale facilities and even then have low count rates. We have also extended the introduction of oxygen K-edge measurements of water to provide more context for this work.

Edits:

p. 2-3: “Soft X-ray second harmonic generation (SXSHG) applies the surface sensitivity of SHG to element-specific X-ray transitions, and can probe electronic valence states not easily accessible with UV-visible spectroscopies⁷⁻¹³. As SXSHG almost-exclusively relies on the intense X-ray pulses produced by X-ray free electron lasers (XFELs), it is a relatively new technique that is still quickly developing. The foundational work on graphite slabs over a range of thicknesses demonstrated the possibility of the measurement and the surface-selectivity of the measurement⁸. A later study on a boronitride-perylene junction showed the sensitivity to buried interfaces as well as the ability to extract intermolecular distances with Angstrom precision⁹. Work on LiNbO₃ has also demonstrated the polarization-dependence of the signal as well as element-dependent signatures from the Li K-edge and Nb N-edge¹⁰.”

p. 3: “The initial states in soft X-ray spectroscopies like SXSHG comprise tightly bound core electrons, which cause the corresponding transitions to be almost entirely dominated by atom-dependent binding energies that are separated by 10s to 100s of eV. In the case of water, soft X-ray absorption spectroscopy (XAS) of the oxygen K-edge is well-established as a sensitive probe of its hydrogen bonding network, and has been the subject of several reviews^{27,28}. This sensitivity arises primarily because of the strong transitions from the core oxygen 1s orbitals and the lowest-lying unoccupied valence orbitals, which have σ^* character and are localized on the hydrogen atoms. The water XAS spectrum is then particularly sensitive to the hydrogen bond donation character of the probed water molecules, and suggests that the SXSHG spectrum can provide similar information for the interfacial water molecules.”

2. Interpretation and Broader Impact:

The manuscript lacks a deeper interpretation of the results and their implications. The conclusion that interfacial water has a different hydrogen-bonding structure than bulk water seems rather expected. As the authors state in the introduction, “Despite the pervasiveness of aqueous interfaces and the many tools developed to study them, there are still substantial open questions about their electronic and molecular structure and how they connect to important macroscopic phenomena.” It would strengthen the paper if the authors explicitly discussed how their results help address these open questions or provide new insights into the link between interfacial molecular structure and macroscopic properties.

We address this simultaneously with point 3 below.

3. Spectral Quality and Data Robustness:

A central result of the study is the SXSHG spectrum shown in Fig. 3C. However, the large error bars and limited number of data points raise concerns about the reliability of the result. As it stands, the spectral quality does not support even semi-quantitative comparisons or analysis. It would be helpful if the authors could provide additional technical and sample replicates to enhance the validity of their findings. Furthermore, a forward-looking discussion on how the spectral quality and signal-to-noise ratio might be improved in future measurements would be valuable. As the field of interfacial water is already subject to considerable debate, limited data quality may risk introducing further inconsistencies rather than resolving them.

As the referee noted, given the complexity of these measurements, one is forced to strike a balance between the robustness of the interpretation and the obtained signal to noise of the data. All XFEL measurements suffer from the inherently stochastic nature of the source at present, ultimately acting as a limit on our ability to make definitive conclusions. The limited availability of this facility time also makes it challenging to get additional experimental replicates. We have made a conscious decision here to not overstate our conclusions given the signal to noise of the experiment. We have also further highlighted the theoretical result that the SXSHG signal is strongly influenced by the H-bond acceptor character of the surface water molecules.

Given all of that, the data appears to support the water structure measured here being consistent with the state of the art in simulations of liquid water, providing support for these computational methods such as machine learned potentials and *ab initio* techniques. In a future work, we will compare and contrast the expected SXSHG spectra of various water models, from DFT/*ab-initio*, machine learned and classical.

We have extended our discussion to include possible future experimental-design improvements and facility upgrades that can improve this field moving forward. As noted below, these measurements can be improved in a straightforward way and additionally provide a broad method to study a broad range of liquids, including solutes and solvents.

Edits:

p. 12-13: “This sensitivity to H-bond acceptors was an unexpected result, as the bulk XAS is known to be more sensitive to H-bond donors.

“While in this current work, the conclusions that we are able to draw are limited due to the low signal-to-noise of the SXSHG feature above the XFEL background, there are a number of strategies that could be pursued to improve the measurement. Selective absorptive and spatial filters can be employed to preferentially remove the XFEL harmonic before reaching the sample target, or a reflective geometry could be employed to minimize the harmonic background by utilizing Brewster’s angle. Target thickness, angle-of-incidence, and X-ray polarization dependence could also be examined in future work to experimentally demonstrate surface selectivity.

“Most significantly, the commissioning of new high repetition rate XFELs such as LCLS-II will enable the collection of SXSHG data at tens of kHz instead of the 120 Hz used in this study. We expect this to help evaluate and validate the various water models/potentials used in the literature for simulating the vapor/water interface. In addition to future measurements on neat water, it will be possible to study a variety of scientifically and industrially interesting systems...”

Reviewer #2 (Remarks on code availability):

I am not an expert in modeling and therefore defer to the evaluation and judgment of other reviewers on this aspect of the work.

Reviewer #3 (Remarks to the Author):

The manuscript innovatively reports soft X-ray second harmonic generation (SXSHG) from the water/vapor interface to elucidate interfacial electronic structure. Key findings reveal distinct SXSHG spectra compared to bulk X-ray absorption, highlighting broken hydrogen-bonding networks at the surface. Combining theoretical calculations, the results indicate that the signal is highly sensitive to single-acceptor H-bond configurations, abundant at the interface. This study opens up new possibilities for studying molecular properties at interfaces using nonlinear X-ray spectroscopy, with significant implications for understanding interfacial chemistry and improving theoretical models of liquid interfaces. Given recent advances in flat liquid sheet microjets and XFEL technology, the study is very timely and significant for advancing surface science.

While the results carry scientific significance and I can recommend the work for publication in principle, a several critical aspects of the manuscript require improvement and clarification to reinforce their conclusions and elevate the manuscript quality.

1. Surface sensitivity

The SXSHG in transmission geometry is employed in this study. As we know that the transmitted sum frequency vibrational spectroscopy and optically SHG is not necessarily surface sensitive due to the bulk quadrupole contribution and longer interaction length. The authors should evaluate the bulk quadrupole contribution in the transmitted SXSHG measurement, which is critical for surface sensitivity on water/vapor of the technique.

Surface sensitivity of our current work: We base the current surface sensitivity claims in the current work based on previous SXSHG measurements and the TD-DFT calculations on both bulk water and the water surface slab. As shown in ED Fig. 7, the simulated SXSHG spectrum of the same system in a 3D bulk configuration is an order of magnitude less intense than that of a 2D slab configuration. This is despite the presence of microscopic heterogeneities in the small water cell (<100 molecules) that would average out in the macroscopic experiment.

Prior SXSHG measurements on graphite targets at the C K-edge found no dependence on the sample thickness on targets between 100 and 700 nm thick within error. The signal from LiOsO₃ at the Li K-edge was seen to disappear below signal to noise when the structure transitioned from noncentrosymmetric to centrosymmetric. The explanation for this is that the technique is

strongly symmetry (and thus interface) selective. Additionally, the polarization dependence of LiNbO_3 is well described by dipole dependence and indeed directly contradicts the idea of a quadrupole dominant measurement.

While not performed for this study, experimental verifications of the surface sensitivity could be potentially performed by adjusting the sample thickness, the sample angle of incidence, or the X-ray polarization. We've included a sentence in the main text to underscore this point.

Quadrupole or higher order contributions: One of the advantages of performing RT-TDDFT in the velocity gauge is that the method provides a full multipole expansion via the applied external vector potential and the resultant time-dependent current density. In this sense, all contributions, dipole, quadrupole, etc., are included, albeit at a nontrivial computational cost.

Edits:

p. 10: "These calculations inherently include bulk-active quadrupole contributions as well (see Materials and Methods). Combined with previous experimental and theoretical work on other samples showing thickness independence⁸ and polarization-dependence¹⁰ consistent with dipole selection rules, these simulations support our claim of surface selectivity."

p. 13: "Target thickness, angle-of-incidence, and X-ray polarization dependence could also be examined in future work to experimentally demonstrate surface selectivity."

p. 29: "One of the advantages of performing RT-TDDFT in the velocity gauge is that the method provides a full multipole expansion via the applied external vector potential and the resultant time-dependent current density. In this sense, all contributions, dipole, quadrupole, etc., are included, albeit at a nontrivial computational cost, meaning the method is also sensitive to bulk-active contributions to the signal."

2. Physical Mechanism Underpinning H-Bond Sensitivity

In the manuscript, the simulation results showed that SXSHG spectra is particularly sensitive to the number of acceptor H-bonds, in contrast to linear XAS measurements, which is sensitive to the degree of donor H-bonds. A deeper physical explanation is needed.

The nature of the sensitivity to hydrogen bonding depends on the localization of the valence electronic orbitals. The low-lying (and most easily characterizable) features in the linear XAS correspond to orbitals localized primarily on the hydrogen atoms, which makes them more sensitive to hydrogen bond donation.

Our core result is that the SXSHG spectra is shifted several eV relative to the static XAS. The valence orbitals here are potentially more localized on the oxygen lone pairs, making them more sensitive to the acceptor hydrogen bonds. One possible explanation is that the excitonic states found by the quantum chemistry calculations found few states spectrally overlapping with the $4a_1$ state that is strongly hydrogen-localized, and instead overlaps with higher energy states.

Edits:

p. 3: “In the case of water, soft X-ray absorption spectroscopy (XAS) of the oxygen K-edge is well-established as a sensitive probe of its hydrogen bonding network, and has been the subject of several reviews^{27,28}. This sensitivity arises primarily because of the strong transitions from the core oxygen 1s orbitals and the lowest-lying unoccupied valence orbitals, which have σ^* character and are localized on the hydrogen atoms. The water XAS spectrum is then particularly sensitive to the hydrogen bond donation character of the probed water molecules, and suggests that the SXXHG spectrum can provide similar information for the interfacial water molecules.”

p. 9: “This stands in contrast to linear XAS measurements, which have been shown to be sensitive to the degree of donor H-bonds due to the low-lying water valence orbitals being localized on the hydrogen atoms,”

p. 11: “These states may provide a partial explanation for the sensitivity to acceptor H-bonds as well, as the $4a_1$ state localized on the hydrogen atoms have poor spectral overlap with the $sx1/sx2$ manifolds, and potentially better overlap with higher energy orbitals with more oxygen character.”

3. Fluence dependent measurement

The fluence dependence measurement data were collected in a range of near ~0 uJ to 100s uJ as shown in figure 3B and the authors mentioned that “the intrinsic shot-to-shot fluctuations of the XFEL pulse intensity were used to generate the fluence dependence for identifying nonlinear signals”. If I did not misunderstand, the pulse energy of every shot is completely random and uncontrollable and fluctuate in a large range shot-to shot? Then, I have a technical puzzle that why the pulse energy of XFEL fluctuate so large? Given that the large energy fluctuation, how will its spectra profile change? This is critical for the spectral resolution.

The fundamental X-ray free electron laser process, spontaneous amplified stimulated emission, is effectively stochastic, which results in large variations in the spectrum and intensity on a per-shot basis. The XLEAP process reshapes the electron beam into a sub-femtosecond current spike, which mitigates most of the spectral variation, however intensity variation is dominated by charge fluctuations in the initial seed beam at the cathode (<http://dx.doi.org/10.1038/s41566-024-01427-w>). The result, over most of the intensity range, is a reasonably consistent spectral profile (an increase in lower photon energies are observed at higher intensities). The low energy pulses, <100 uJ are less consistent but also less significant for the analysis. The intensity-dependent spectral profiles were used in the model fitting of the SHG and TA signals as detailed in the methods section.

Edits:

A histogram of the GMD values and the average fundamental and harmonic pulse spectra per histogram bin were added to Extended Data Fig. 1.

p. 6: “The pulse energy varies stochastically from <100 to >500 mJ with moderate spectral changes due to shot-by-shot variation in the electron beam³¹.”

p. 15: “Histograms of pulse energies as measured by the GMD for six consecutive runs are shown in Extended Data Fig. 1D, showing the machine behavior is consistent over these timescales. Average spectra of the fundamental and harmonic for each GMD bin are shown in Extended Data Fig. 1E and 1F, respectively. There are moderate changes to the spectrum with pulse energy, especially in the lowest quartile, but the higher energy pulses are fairly consistent. The full analysis described below uses these intensity-dependent spectra in the calculation of the TA and SHG contributions to the signal.”

ED Fig 1. Caption: “**D.** Histogram of GMD pulse energy readings across six separate experimental runs. Baseline-subtracted jet-out fundamental (**E**) and harmonic (**F**) spectra binned to GMD pulse energy.”

One more small suggestion: it would be better to put a sentence stating GMD energy measurements are performed in situ and do not influence the shot energy. This will be friendly for the readers who are not the X-ray field.

We have added the additional clarification that the reviewer suggested.

Edit:

p. 5: “The pulse energy was measured non-invasively shot-by-shot using a gas monitor detector (GMD), which we will use as a proxy for intensity throughout this work.”

Fig. 1 Caption: “The pulse energy is measured non-invasively by a nitrogen gas attenuator (GMD).”

Minor comments:

1. There are dual red ribbons in figure 2d. What’s the origination of the other one?

The upper red ribbon is likely a background subtraction artifact, since it does not seem to reproduce in the other covariance difference maps as shown in Extended Data Fig. 4

2. In the text, the authors use the descriptions of fundamental fluence, intensity and pulse energy in the fluence dependence section. I believe they express the same thing. In figure 3b, it’s better to use pulse energy and uJ for x-axis, so it can be directly related to figure 3a.

We agree with the reviewer that we can unify these terms in the text. We use “intensity” throughout most of the text to try and avoid confusion with “photon energy”. We use “pulse energy” when plotting figures, as this is what we are directly measuring with the GMD. We added the clarification that we are using pulse energy as a proxy for intensity shown above.

Edits:

We have changed the scale of Fig. 3B (and corresponding panels in Extended Data Fig. 4) to match pulse energy per the reviewer’s recommendation. The x-axes labels there and in ED Fig. 2 were relabeled as “Pulse Energy”.

Throughout the text, “fluence” -> “intensity”

Moreover, the data in figure 3b seems not be the same with what showed in extended figure 4 at 550eV (where the error-bars are obviously not the same.) Please check the data and specify the difference.

The data shown in Extended Data Fig. 4 is from a different run at that photon energy. (I.e., the second point at 550 eV in Fig. 3C), and to avoid redundancy in figures. We added clarification in the text.

Edit:

ED Fig. 4 caption: “The dataset shown for 550 eV is from a separate set of runs from the dataset shown in Figs. 2 and 3.”

3. In page 8, line 186, the description of “intense main peak at 542.5 eV” is not in line with the figure 4c for A1D0, where the peak is at 544 eV. The authors should check the consistency.

We have corrected this error in the text

Edit:

p. 9: “In particular, interfacial single acceptor/no donor species (A_1D_0 , blue trace in Fig. 4C) presents several SXSHG features at ~ 532 eV, 540 eV and a particularly intense main peak at 544 eV.”

4. What’s the chamber pressure for this SXSHG measurement? Please specify this parameter in the methods.

The chamber pressure is approximately 1 mTorr when the jet is running. We have added this to the methods section.

Edit:

p. 14: “Measurements were made in-vacuum with a chamber pressure of ~ 1 mTorr.”